# Masked Autoencoders are PDE Learners

**Anthony Zhou**                                                                *ayz2@andrew.cmu.edu*
*Department of Mechanical Engineering*
*Carnegie Mellon University*

**Amir Barati Farimani**                                                        *barati@cmu.edu*
*Department of Mechanical Engineering*
*Department of Machine Learning*
*Carnegie Mellon University*

**Reviewed on OpenReview:** *https://openreview.net/forum?id=rZNuiFwXVs*

## Abstract

Neural solvers for partial differential equations (PDEs) have great potential to generate fast and accurate physics solutions, yet their practicality is currently limited by their generalizability. PDEs evolve over broad scales and exhibit diverse behaviors; predicting these phenomena will require learning representations across a wide variety of inputs which may encompass different coefficients, boundary conditions, resolutions, or even equations. As a step towards generalizable PDE modeling, we adapt masked pretraining for physics problems. Through self-supervised learning across PDEs, masked autoencoders can consolidate heterogeneous physics to learn rich latent representations. We show that learned representations can generalize to a limited set of unseen equations or parameters and are meaningful enough to regress PDE coefficients or the classify PDE features. Furthermore, conditioning neural solvers on learned latent representations can improve time-stepping and super-resolution performance across a variety of coefficients, discretizations, or boundary conditions, as well as on certain unseen PDEs. We hope that masked pretraining can emerge as a unifying method across large, unlabeled, and heterogeneous datasets to learn latent physics at scale.

## 1   Introduction

The physical world is incredibly complex; physical phenomena can be extremely diverse and span wide spatiotemporal scales—from neuron excitations to turbulent flow to even global climate. Importantly, many of these phenomena can be mathematically modeled with time-dependent partial differential equations (PDEs)(FitzHugh, 1961; Nagumo et al., 1962; Lorenz, 1963). These PDEs are generally analytically intractable and require the use of numerical solvers to obtain approximate solutions. For complex physics, these solutions can often be slow to obtain; furthermore, different PDEs often require a careful design of tailored solvers.

Advances in deep learning have led to the design of a new class of solvers for PDEs. These neural solvers can be extremely fast and display resolution invariance; however, neural networks introduce training difficulties and a lack of theoretical guarantees. Many important advances have been made to address these challenges, with models becoming faster than numerical solvers within well-studied PDEs under certain setups, proposing error bounds, and being extended to solve real-world problems. (Raissi et al., 2019; Lu et al., 2019; Li et al., 2020; Cao, 2021; Brandstetter et al., 2022; Li et al., 2023; Kovachki et al., 2021; Li et al., 2024).

A current frontier in neural PDE solvers lies in generalizing solvers to different parameters, conditions, or equations, thereby avoiding the need to collect new data and retrain networks when given unseen PDE dynamics. Prior work in this space has explored many methods to achieve this, from directly conditioning on PDE coefficients (Takamoto et al., 2023; Lorsung et al., 2024; Shen et al., 2024) to pretraining foundation models across various equations (Subramanian et al., 2023; McCabe et al., 2023; Hao et al., 2024). Despite

these advances, generalizable neural solvers remain a significant challenge. PDEs can be incredibly diverse, and neural solvers must adapt to different coefficients, geometries, discretizations, or boundary conditions.

As a step towards addressing generalizability, we propose adapting masked pretraining methods to the PDE domain. This is motivated by the observation that masked pretraining can learn highly meaningful and broad knowledge in the computer vision and language domains (Devlin et al., 2018). In addition, masked modeling approaches are known to scale well to large and diverse datasets (He et al., 2021). In practice, masked pretraining can also be easily implemented since it makes no prior assumptions about the PDE data, and modern training pipelines and masked modeling architectures are efficient (Vaswani et al., 2023). Lastly, it is hypothesized that masked modeling approaches can generalize well due to having very low inductive bias from masking large portions of the data (Feichtenhofer et al.).

As such, to study masked modeling within the PDE domain, we train masked autoencoders on a diverse set of 1D and 2D PDE data and evaluate their learned representations. We demonstrate that self-supervised masked pretraining can learn latent structure that can express different coefficients, discretizations, boundary conditions or PDEs under a common representation. Furthermore, we show that masked autoencoders (MAEs) can learn highly structured latent spaces through masking alone. MAE models can be used to improve downstream tasks such as predicting PDE features or guiding neural solvers in time-stepping or super-resolution through providing meaningful context. Our contributions suggest the possibility to transfer the scalability and flexibility of masked modeling from language and vision domains to physics—creating rich, unified representations of diverse physics through self-supervised learning. We provide the code and datasets used in this study here: `https://github.com/anthonyzhou-1/mae-pdes` .

## 2 Related Work

**Neural PDE Solvers** The field of neural PDE solvers has grown rapidly and has shown great advances in both the accuracy of solutions and the ability to adapt to PDE parameters. Infinite-dimensional neural operators (Li et al., 2020; Kovachki et al., 2023; Lu et al., 2019) have shown impressive accuracy in solving time-dependent PDEs by learning the mappings between initial conditions and solutions. However, these methods alone have shown brittleness with respect to changing PDE coefficients or boundary conditions (Gupta and Brandstetter, 2022; Lu et al., 2021), prompting recent work to allow neural solvers to adapt to different PDE conditions.

A variety of approaches have considered adding PDE dynamics information or time-dependent trends to neural solvers. Common neural solvers can support conditional prediction through architecture choices (Gupta and Brandstetter, 2022), and novel architectures can be designed to explicitly operate with PDE parameter knowledge (Brandstetter et al., 2022). Beyond directly conditioning on PDE dynamics, a class of neural PDE solvers has proposed the addition of an encoder or adaptive network to inform a forecaster network of different PDE coefficients (Wang et al., 2021; Kirchmeyer et al.; Takamoto et al., 2023; Lorsung et al., 2024). At an even higher level, meta-learning approaches have been adapted to PDE learning to maximize shared learning across different physics (Yin et al., 2021; Zhang et al., 2023).

**Pretraining for PDEs** As an effort to work towards more generalizable PDE neural solvers, recent work has followed the success of pretraining and foundational models in the broader deep learning community. Based on contrastive pretraining methods in computer vision problems, (Chen et al., 2020; Schroff et al., 2015; Zbontar et al., 2021; Bardes et al., 2022), contrastive PDE methods aim to leverage equation coefficients (Lorsung and Farimani, 2024), physical invariances (Zhang et al., 2023), or Lie point symmetries (Mialon et al., 2023; Brandstetter et al., 2022) to define differences in PDE dynamics that can be organized in a latent space. Another approach in PDE pretraining follows observed in-context learning and emergent behavior in LLMs (Wei et al., 2022; Brown et al., 2020; Radford et al.) to design neural PDE solvers that are capable of following prompted PDE examples to forecast unseen dynamics (Yang et al., 2023; Chen et al., 2024).

A more straightforward pretraining method focuses on directly training neural solvers to transfer to new PDE dynamics (Goswami et al., 2022; Chakraborty et al., 2022; Wang et al., 2022). This approach has also been scaled by training neural solvers with large and diverse training sets to characterize its transfer behavior (Subramanian et al., 2023), as well as shown to be generally more effective over other pretraining

strategies (Zhou et al., 2024). As a step toward large-scale modeling, more principled training approaches have been proposed to learn PDE dynamics across diverse physics at scale. Recent work has proposed a combinatorial neural operator that learns different dynamics as separate modules (Tripura and Chakraborty, 2023), embedding separate PDEs to a common space to do multi-physics prediction (McCabe et al., 2023), incorporating denoising with scalable transformer architectures while training across diverse PDE datasets (Hao et al., 2024), and using a unified PDE embedding to align LLMs across PDE families (Shen et al., 2024).

**Masked Pretraining** Masked reconstruction is a popular technique popularized by the language processing (Devlin et al., 2018) and vision (Dosovitskiy et al., 2020; Xie et al., 2021; He et al., 2021) domains to pretrain models for downstream tasks. Masked modeling is a broad field that spans many masking strategies, architectures, and applications (Li et al., 2024); this ubiquity is attributed to scalability and architecture breakthroughs (Vaswani et al., 2023) that allow meaningful context to be learned through masked reconstruction (Cao et al., 2022). In the field of neural PDE solvers, masked pretraining has initially been explored as a method to pretrain neural solvers directly (Chen et al., 2024). However, we take a separate approach by investigating if models can understand physics through masked self-supervised learning and how these latent representations are manifested. After validating this learning method, we seek to understand if this knowledge can be used to benefit common PDE tasks.

**Situating our Contribution** To frame our contribution within these past works, we draw comparisons with broader deep learning efforts to pretrain large encoders through self-supervision, such as BERT (Devlin et al., 2018) or CLIP (Radford et al., 2021; Ramesh et al., 2022), to be used in downstream tasks. For many reasons (e.g. data availability, architectures, pretraining strategies, compute resources), an equivalent work does not currently exist for PDEs. However, we hope to advance this research thrust to train general-purpose physics encoders, which could be of benefit to the PDE community. Specifically, pretrained encoders are often leveraged to accelerate model development and training by outsourcing certain architecture modules to pretrained models. Furthermore, having standardized, encoded representations of diverse PDEs can help in interfacing with conventional ML pipelines, whether it be allowing LLMs to interface with PDE data or diffusion backbones to condition on PDE inputs.

## 3 Methods

We describe our main methods in Figure 1. We first pretrain an encoder and decoder to reconstruct masked inputs. This pretraining objective can have a few benefits. Masking destroys inherent biases present in the data; especially at high masking ratios, this forces models to learn very general latent representations. Lastly, masked modeling does not assume any prior physics or leverage features specific to a particular PDE, making the approach applicable to a wide variety of physics.

After pretraining, we evaluate using the learned representations for three downstream tasks after discarding the decoder. The first task is to regress or classify PDE features, such as predicting coefficient values or boundary conditions of a PDE sample. This can be done by directly appending a linear head to the pretrained encoder. The second task is to improve the performance of neural solvers by conditioning on an encoded representation of input physics. Both the encoded and original representations of the data are passed to the neural solver. Lastly, we consider improving super-resolution performance by passing an up-sampled input to a neural solver and conditioning on an encoded representation of the low-resolution PDE input.

### 3.1 Masked Pretraining for PDEs

We adapt the popular Masked Autoencoder (MAE) approach (He et al., 2021; Xie et al., 2021; Feichtenhofer et al.) to train ViT models (Dosovitskiy et al., 2020; Arnab et al., 2021) to reconstruct 1D and 2D PDE data. Data are partitioned into non-overlapping patches before a random subset of these patches is sampled to be masked. The masked patches are omitted from the encoder input; the encoder embeds only the unmasked patches through a series of transformer blocks, which allows for larger encoders and faster training at large masking ratios (He et al., 2021). The encoded patches are then recombined with mask tokens according to their position in the PDE solution. Positional embeddings are added again to preserve positional information

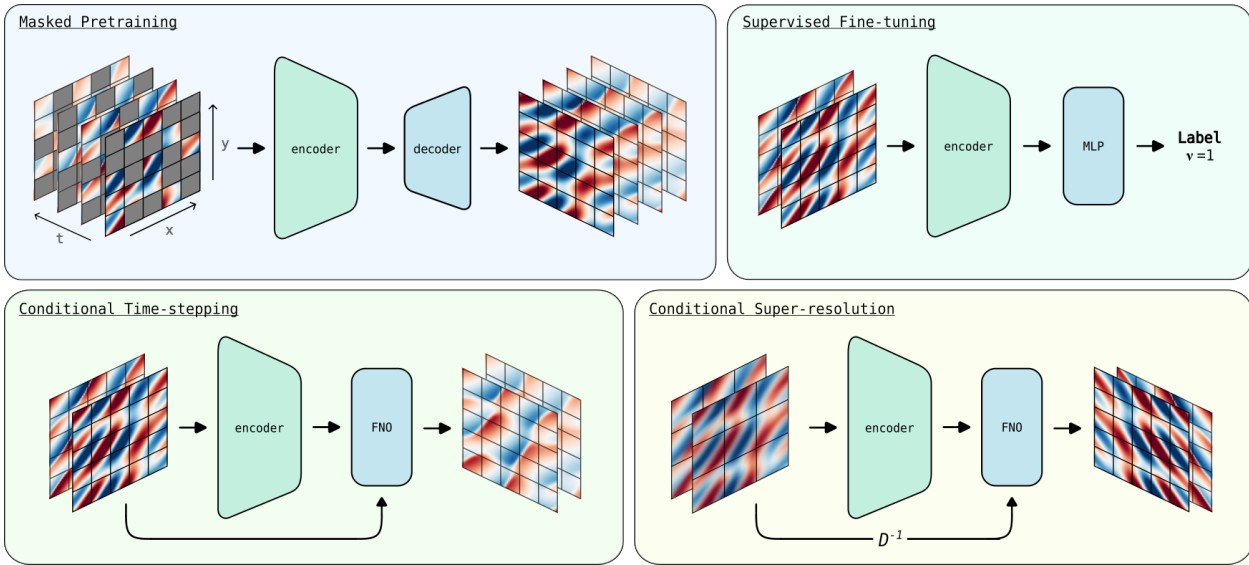

**Figure 1:** We investigate learning diverse PDE dynamics with masked autoencoders (MAE) and using learned representations to benefit various downstream tasks. (*Masked Pretraining*) An encoder is trained on unmasked patches of spatiotemporal PDE data, while a decoder reconstructs true data from latent encodings and learned mask tokens. (*Supervised Fine-tuning*) Pretrained encoders can be used to quickly regress equation coefficients or predict key PDE features. (*Conditional Time-stepping*) Neural solvers can achieve higher accuracy predictions through conditioning on MAE encodings. (*Conditional Super-resolution*) SR models can also benefit from conditioning on MAE encodings, using a discretization inversion ($D^{-1}$) and neural operator (Yang et al., 2023) to predict high-resolution physics.

before the latent encodings and mask tokens are decoded. The decoder can be shallower and narrower than the encoder because it is discarded in downstream tasks (He et al., 2021), which further reduces training costs. The decoded tokens are projected into the PDE space through a linear layer before reshaping the patches to the input dimension. Lastly, the output is compared to ground truth PDE data through a MSE loss.

Within this general framework, the primary considerations affecting performance and compute are model size, masking ratio, and patch size. We study the effects of these parameters on the resulting model's reconstruction performance. In general, we find that increasing model size requires more data to be effective. Furthermore, increased masking ratios decrease the reconstruction performance but may lead to better learned representations. Lastly, decreasing patch size improves reconstruction performance at the cost of increased compute. For additional details and hyperparameters we direct readers to Appendix A.

### 3.2 Lie Point Symmetry Data Augmentation

To emulate a larger pretraining dataset and aid in generalization, we investigate using Lie augmentations during pretraining, an approach that follows the use of data augmentations in vision or video pretraining (He et al., 2021; Xie et al., 2021; Feichtenhofer et al.). Given a PDE, one can derive its Lie symmetries as a set of transformations $\{g_1, \ldots, g_i\}$, each with a variable $\epsilon_i$ that modulates the magnitude of the transformation. At training time, we apply $g_i$ sequentially, each with a randomly sampled $\epsilon_i$ to randomly augment PDE samples. This augmented PDE sample could represent a solution that has been shifted in space, time, or magnitude, among other transformations, but still propagates dynamics according to the original PDE. For a more detailed discussion of Lie point symmetries for PDEs, we refer the reader to Olver (1986) and Mialon et al. (2023). To understand the effects of Lie data augmentation on reconstruction accuracy, we apply randomly

**Table 1:** Error at different augmentation probabilities

| Aug. Prob. | Error |
|---|---|
| 0.00 | 2.48e-03 |
| 0.25 | 1.24e-03 |
| 0.50 | 1.17e-03 |
| 0.75 | 1.27e-03 |
| 1.00 | 1.37e-03 |

augment training samples with different probabilities, with 0 representing no augmentation and 1 representing always augmenting samples. The resulting validation MSE reconstruction losses are given in Table 1 after training on 1D KdV-Burgers data. We find that over-augmenting samples may shift the training distribution too much; furthermore, we generally find that the best results use smaller augmentation magnitudes $\epsilon$.

**Multi-Resolution Pretraining**  PDE data are often discretized on a mesh, which can be both irregular and of different resolutions. This can present a challenge for vanilla transformers, which use the same positional encodings regardless of variations in input sequence length. This is especially pronounced in 2D; flattening two spatial dimensions results in positional information needing to wrap around inputs, and as a result, the same positional embedding can represent very different points in space when given different discretizations. Indeed, many approaches have been proposed to adapt ViT models to inputs of varying resolutions or patch sizes (Beyer et al., 2023; Tian et al., 2023; Dehghani et al., 2023; Fan et al., 2024). However, to avoid changes to the ViT backbone, we consider different strategies to accommodate varying input lengths. In particular, we consider the setup where inputs of shape $(n_t, n_x)$ or $(n_t, n_x, n_y)$ can vary along their spatial dimension $(n_x, n_y)$ by multiples of the patch size $(p_x, p_y)$. Snapshots in time, or time windows, are sampled with length $n_t$, which discretizes the PDE trajectories to a common time dimension.

The simplest strategy would be to pad the inputs to the maximum sequence length (*Pad*). However, this does not address the fact that positional information becomes overloaded and only provides information about input length. To address this, we consider interpolating positional embeddings from the maximum sequence length to variable sequence lengths (*Interp.*). This would allow tokens to receive approximate positions; however, this would not be accurate for irregular grids. Lastly, to adapt to arbitrary grids, we consider using an embedder network, which could be a 1D or 2D CNN (depending on the spatial dimension), to project the spatial coordinate grid to a token (*Token*). We compare the validation MSE reconstruction loss of using different strategies in Table 2 after training on

**Table 2:** Error w/ different resolution strategies

| Strategy | Error |
|----------|----------|
| None | 4.40e-03 |
| Pad | 3.81e-03 |
| Interp. | 3.23e-03 |
| Token | 1.89e-03 |

multiresolution KdV-Burgers data, observing consistent improvement by using more sophisticated methods for embedding spatial discretizations. In 1D, multiresolution pretraining is significantly easier to learn, so we use the *Interp.* strategy for simplicity. In 2D, the use of tokenized spatial grid information becomes more important, so we use the *Token* strategy for 2D multiresolution experiments.

## 4 Experimental Setup

### 4.1 PDEs and Datasets

We describe a variety of PDEs used for masked pretraining and downstream evaluation. In 1D, we pretrain MAE models on the KdV-Burgers equation only, while in 2D we pretrain on the Heat, Advection, and Burgers equations simultaneously. In all PDEs, coefficients and forcing terms are randomly sampled to produce diverse dynamics within a dataset.

**1D KdV-Burgers Equation:** The KdV-Burgers equation (KdV-B) contains the Heat, Burgers, KdV equations as corner cases modulated by coefficients $(\alpha, \beta, \gamma)$ (Brandstetter et al., 2022; Jeffrey and Mohamad, 1991):

$$\partial_t u + \alpha u \partial_x u - \beta \partial_{xx} u + \gamma \partial_{xxx} u = \delta(t, x) \tag{KdV-B}$$

Both initial conditions and forcing terms are generated from the periodic function $\delta(t, x)$, where we uniformly sample $A_j \in [-0.5, 0.5], \omega_j \in [-0.4, 0.4], l_j \in \{1, 2, 3\}, \phi_j \in [0, 2\pi)$ while fixing $J = 5, L = 16$.

$$\delta(t, x) = \sum_{j=1}^{J} A_j sin(\omega_j t + 2\pi l_j x/L + \phi_j), \quad u(0, x) = \delta(0, x) \tag{1}$$

To generate diverse PDE samples, the coefficients are sampled uniformly from $\alpha \in [0, 3], \beta \in [0, 0.4], \gamma \in [0, 1]$. Furthermore, samples are generated with a discretization $(n_t, n_x) = (250, 100)$ on an interval $x = [0, 16]$

from $t = 0$ to $t = 2$. For 1D multi-resolution experiments, PDE data from the KdV-Burgers equation is downsampled to variable spatial resolutions.

**1D Heat and Burgers Equations:** The 1D Heat and inviscid Burgers (Brandstetter et al., 2022) equations are used to evaluate performance on PDEs that are a subset of pretraining samples. Furthermore, to evaluate extrapolation to unseen boundary conditions (BCs), samples of the Heat equation are also generated with Dirichlet and Neumann BCs in addition to periodic BCs.

$$\partial_t u - \nu \partial_{xx} u = \delta(t, x) \tag{Heat}$$

$$\partial_t u + u \partial_x u = \delta(t, x) \tag{Burgers}$$

We solve the equations with the same periodic BCs, initial conditions, and forcing function setup and as the 1D KdV-Burgers equation, but by setting the appropriate coefficient values. Specifically, we uniformly sample $\nu = \beta \in [0.1, 0.8]$ for the Heat equation and fix $\alpha = 0.5, \beta = 0$ to model the inviscid Burgers equation. These equations are also solved with a discretization $(n_t, n_x) = (250, 100)$ on an interval $x = [0, 16]$ from $t = 0$ to $t = 2$. To generate data for the Heat equation that enforces Dirichlet or Neumann boundary conditions, we write the Heat equation in its variational form after an implicit Euler discretization:

$$\int_\Omega (uv + \Delta t \nabla u \cdot \nabla v) dx = \int_\Omega (u^n + \Delta t f^{n+1}) v dx \tag{2}$$

This formulation can be solved using FEniCS (Alnaes et al., 2015; Logg et al., 2012). To simplify the boundary value problem, we set the forcing term $f = 0$. Furthermore, we set $[u(x = 0) = u(x = L) = 0]$ to enforce Dirichlet BCs, and $[\partial_x u(x = 0) = \partial_x u(x = L) = 0]$ to enforce Neumann BCs; however, in both of these cases, the initial conditions in Equation 1 need to be modified to respect the new BCs.

$$u(0, x) = \sum_{j=1}^{J} A_j sin(2\pi l_j x/L + \phi_j), \quad \text{Dirichlet ICs} \tag{3}$$

$$u(0, x) = \sum_{j=1}^{J} A_j cos(2\pi l_j x/L + \phi_j), \quad \text{Neumann ICs} \tag{4}$$

In both equations, we uniformly sample $A_j \in [-0.5, 0.5], l_j \in \{1, 2, 3\}, \phi_j \in \{0, \pi\}$ while fixing $J = 5, L = 16$.

**1D Advection, Wave, and Kuramoto-Sivashinsky Equations:** The Advection (Adv), Wave, and parameter-dependent Kuramoto-Sivashinsky (KS) (Lippe et al., 2023) equations are considered to evaluate downstream performance to new equations; the equations contain PDE terms that are unseen during pretraining. Additionally, the Wave equation is generated with Dirichlet and Neumann BCs (Brandstetter et al., 2022) to evaluate unseen BCs on novel PDE dynamics.

$$\partial_t u + c \partial_x u = 0 \tag{Adv}$$

$$\partial_{tt} u - c^2 \partial_{xx} u = 0 \tag{Wave}$$

$$\partial_t u + u \partial_x u + \nu \partial_{xx} u + \partial_{xxxx} u = 0 \tag{KS}$$

For the 1D Advection equation, initial conditions are generated according to Equation 1, the wave speed is uniformly sampled from $c \in [0.1, 5]$, and periodic BCs are used. The solution domain and discretization are the same as previous cases, with $(n_t, n_x) = (250, 100)$, $x = [0, 16]$, and time ranging from $t = 0$ to $t = 2$.

For the 1D Wave equation, we solve with Dirichlet ($u(x = 0) = u(x = L) = 0$) and Neumann ($\partial_x u(x = 0) = \partial_x u(x = L) = 0$) BCs, resulting in waves that either bounce or reflect off boundaries. The wave speed is fixed at $c = 2$, and the initial condition is a Gaussian pulse with unit amplitude and with its peak randomly sampled on the spatial domain. Lastly, the equation is solved from $t = 0$ to $t = 100$ on the interval $x = [-8, 8]$ with a discretization $(n_t, n_x) = (250, 100)$.

For the 1D KS equation, we use periodic BCs with initial conditions from Equation 1. Following the data setup proposed by Lippe et al. (2023), we additionally uniformly sample $\nu \in [0.75, 1.25]$ to vary the second-order

term in the KS equation. Furthermore, due to the unique dynamics of the KS equation, we solve the PDE from $t = 0$ to $t = 100$ on the interval $x = [0, 64]$ with a discretization of $(n_t, n_x) = (100, 100)$.

**2D Heat, Advection, and Burgers Equations:** The 2D Heat, Advection (Adv), and scalar Burgers (Rosofsky et al., 2023) equations are considered for both pretraining and downstream evaluation. For 2D multi-resolution experiments, data from these equations are downsampled to variable spatial resolutions.

$$\partial_t u - \nu \nabla^2 u = 0 \tag{Heat}$$
$$\partial_t u + \mathbf{c} \cdot \nabla u = 0 \tag{Adv}$$
$$\partial_t u + u(\mathbf{c} \cdot \nabla u) - \nu \nabla^2 u = 0 \tag{Burgers}$$

For the Heat equation, we uniformly sample the $\nu \in [2 \times 10^{-3}, 2 \times 10^{-2}$; for the Advection equation, we uniformly sample $\mathbf{c} = [c_x, c_y] \in [0.1, 2.5]^2$; and for the Burgers equation, we uniformly sample $\nu \in [7.5 \times 10^{-3}, 1.5 \times 10^{-2}$, and $\mathbf{c} = [c_x, c_y] \in [0.5, 1.0]^2$. For all equations, we use periodic BCs and solve on a grid $(n_t, n_x, n_y) = (100, 64, 64)$ on a solution domain $(x, y) = [-1, 1]^2$ from $t = 0$ to $t = 2$. Lastly, initial conditions are generated from:

$$u(0, x, y) = \sum_{j=1}^{J} A_j \sin(2\pi l_{xj} x / L + 2\pi l_{yj} y / L + \phi_j) \tag{5}$$

Initial condition parameters are uniformly sampled from $A_j \in [-0.5, 0.5], \omega_j \in [-0.4, 0.4], l_{xj} \in \{1, 2, 3\}, l_{yj} \in \{1, 2, 3\}, \phi_j \in [0, 2\pi)$ while fixing $J = 5, L = 2$.

**2D Navier-Stokes Equations:** Following the setup from Li et al. (2020), we consider the incompressible Navier-Stokes (NS) equations in vorticity form, but randomly sample the viscosity $\nu$ and forcing function $f(x)$ amplitude. To ensure consistency with the pretraining dataset, our experiments model NS dynamics as a scalar vorticity field; from this the velocity field can be derived from the Biot-Savart Law.

$$\partial_t \omega + u \cdot \nabla \omega - \nu \nabla^2 \omega = f(x), \quad \nabla \cdot u = 0 \tag{NS}$$

The solution is solved on a grid $(n_t, n_x, n_y) = (100, 64, 64)$ on a solution domain $(x, y) = [0, 1]^2$ from $t = 0$ to $t = 25$. PDE parameters are uniformly sampled from $\nu \in \{\{1, 2, 3, 4, 5, 6, 7, 8, 9\} \times 10^{-\{6,7,8,9\}}\}$ and $A \in \{1, 2, 3, 4, 5, 6, 7, 8, 9, 10\} \times 10^{-3}$. Lastly, initial conditions $\omega_0$ are sampled according to Li et al. (2020) from a Gaussian random field.

## 4.2 Data Augmentations

We implement Lie Point Symmetry Data Augmentations according to Brandstetter et al. (2022), including shifting and resampling PDE solutions with the Fourier shift theorem. Since we only augment PDE samples during pretraining, we consider symmetry groups for the 1D KdV-Burgers equation, as well as the 2D Heat, Advection, and Burgers equations. The 1D KdV-Burgers equation has the following Lie subalgebras (Ibragimov, 1993):

$$X_1 = \frac{\partial}{\partial t}, \quad X_2 = \frac{\partial}{\partial x}, \quad X_3 = \alpha t \frac{\partial}{\partial x} + \frac{\partial}{\partial u} \tag{6}$$

Taking the exponential map results in the following Lie groups (Ibragimov, 1993):

$$g_1(\epsilon)(x, t, u) = (x, t + \epsilon, u), \quad \text{(Time Shift)} \tag{7}$$
$$g_2(\epsilon)(x, t, u) = (x + \epsilon, t, u), \quad \text{(Space Shift)} \tag{8}$$
$$g_3(\epsilon)(x, t, u) = (x + \epsilon t, t, u + \epsilon), \quad \text{(Galilean Boost)} \tag{9}$$

For the 2D Heat, Advection, and Burgers equations, there are many possible Lie subalgebras (Ibragimov, 1993). For simplicity, we only consider a basic subset of these that apply to all three equations, however, there is ample room to implement more symmetries:

$$X_1 = \frac{\partial}{\partial t}, \quad X_2 = \frac{\partial}{\partial x}, \quad X_3 = \frac{\partial}{\partial y}, \tag{10}$$

These result in the following Lie groups in 2D:

$$g_1(\epsilon)(x, y, t, u) = (x, y, t + \epsilon, u), \quad \text{(Time Shift)} \tag{11}$$
$$g_2(\epsilon)(x, y, t, u) = (x + \epsilon, y, t, u), \quad \text{(X Shift)} \tag{12}$$
$$g_3(\epsilon)(x, y, t, u) = (x, y + \epsilon, t, u), \quad \text{(Y Shift)} \tag{13}$$

## 5 Results

In this section, we provide the results of different experiments designed to understand the capabilities of masked autoencoders. Firstly, we would like to understand the reconstruction capabilities of the masked autoencoder, or if the pretraining goal given by the masked objective is being met. Once MAE performance is validated on its pretraining objective, we seek to understand and visualize the representations learned by the MAE during pretraining. This can be done by projecting the latent embeddings to a lower dimension to visualize qualitative trends. Although insightful, another more rigorous evaluation is the regression/classification of physical variables. These are easy quantities to derive and are usually trivially known *a priori*, however, they can serve as a probe to quantitatively gauge model knowledge rather than rely on qualitative latent trends. This is analogous to a linear probe used to predict image rotations or grayscale vs. color in self-supervised learning for computer vision (Chen et al., 2020); indeed, coefficient regression has been used in prior PDE literature to gauge model performance after self-supervised pretraining (Mialon et al., 2023).

While these three tasks (masked reconstruction, latent visualization, variable regression/classification) can provide knowledge, they are generally not useful on their own. To extend masked pretraining to practical tasks, we consider using a pretrained encoder to improve PDE time-stepping or super-resolution. In particular, we are interested in whether the representations learned during masked pretraining can help in diverse physics scenarios by providing additional context to neural PDE solvers during time-stepping or super-resolution.

### 5.1 MAE Pretraining

MAE models are trained on 10000 samples of 1D KdV-Burgers PDE data in 1D and 12288 samples of 2D Heat, Advection, and Burgers PDE data in 2D. We display example results from masked pretraining in Figures 2 and 3. A notable difference from vision and video domains is that physics does not follow human-recognizable structure or descriptions (e.g. backgrounds, actions, faces, shapes, etc.); furthermore, in addition to the overall meaning, the numerical accuracy of the reconstruction is important. Despite this, MAE models are able to capture underlying physics and reconstruct PDEs within the pretraining set well—both in 1D and 2D, and at high masking ratios (75% and 90%). In general, for PDEs that are similar to those seen during pretraining, such as the 1D Heat or inviscid Burgers equation, MAE models tend to interpolate well. Furthermore, given information about the spatial discretization, MAE models can adapt to different PDE resolutions when trained to reconstruct multi-resolution inputs. This is true in 2D as well, with example results shown in Appendix B.2.

For PDEs that contain novel equation terms or boundary conditions (BCs), the MAE extrapolation performance is limited. Zero-shot reconstruction of the 1D Advection and KS equations shows mild trends, while reconstruction of the 2D Navier-Stokes equations is ineffective. To address this gap and investigate whether MAE models can perform on complex high-resolution physics with multiple variables, we train an MAE model to reconstruct pressure and velocity on 2D smoke buoyancy data with varying buoyancy factors (Gupta and Brandstetter, 2022) and qualitatively show its reconstruction in Appendix B.3. In general, MAE models can adapt to complex scenarios and multiple physical variables; however, many of the fine details (e.g. eddies, shedding) become lost at high masking ratios.

Since the 1D KdV-Burgers equation is solved with periodic BCs, we evaluate MAE extrapolation to Dirichlet and Neumann BCs when reconstructing the 1D Heat and Wave equations, shown in Appendix B.1. Overall

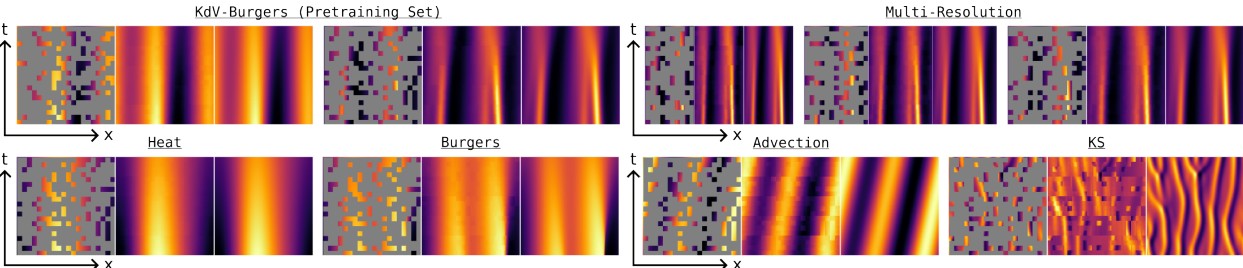

**Figure 2:** Example results after training on the 1D KdV-Burgers equation with a masking ratio of 75%. For each triplet, we show the masked PDE (left), the MAE reconstruction (middle), and the ground-truth (right), and plot space and time on the $x$ and $y$ axes respectively. The MAE can reconstruct multiple resolutions of KdV-Burgers data and interpolate to the 1D Heat and inviscid Burgers equations. For the 1D Advection and KS equations, which contain novel PDE terms ($u_x, u_{xxxx}$), the extrapolation performance is limited.

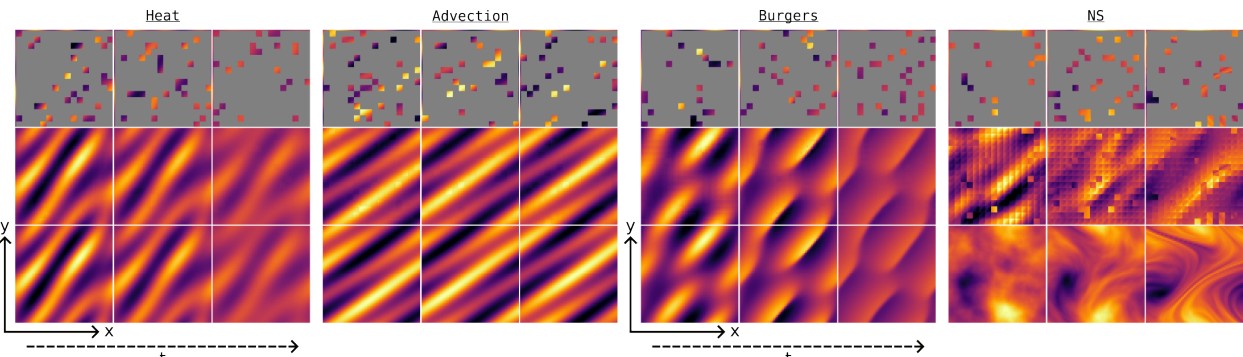

**Figure 3:** Example results after training on a combined set of the 2D Heat, Advection, and Burgers equations with a masking ratio of 90%. We show the masked PDE (top), the MAE reconstruction (middle), and the ground truth (bottom), plotting space on the $x$-$y$ axis at multiple snapshots in time. Despite good performance within the training set, the model is unable to extrapolate to the Navier-Stokes (NS) equations, which contain novel initial conditions, forcing terms, and dynamics.

trends remain consistent; the Heat equation, being more similar to the KdV-Burgers equation, shows limited reconstruction performance when extrapolating to novel BCs, while the Wave equation introduces a novel PDE term ($u_{tt}$) and initial condition (Gaussian pulse), and as a result the zero-shot reconstruction is poor.

## 5.2  Latent Space Evaluation

To better understand the latent representation learned by masked pretraining, we use the MAE encoder to embed various PDE validation samples and visualize embeddings with t-SNE (van der Maaten and Hinton, 2008) in Figure 4. Through self-supervised learning, MAE models can learn trends in PDEs without labeled data and with limited extrapolation abilities. For example, through masked reconstruction of the 1D KdV-Burgers equation, MAE models can learn coefficient-dependent trends (Figure 4**A**). This can be applied to PDEs not seen during training, as the same MAE model can distinguish samples from the Advection equation based on wave speed $c$ (Figure 4**B**). This extrapolation is also observed when embedding samples from different PDEs; representations learned from pretraining on the KdV-Burgers equations allow models to cluster PDE samples from the Heat, Burgers, Advection, and KS equations (Figure 4**C**). Lastly, MAE models are able to distinguish varying resolutions of PDE data after multi-resolution training, suggesting high model capacity across both diverse PDEs and discretizations (Figure 4**D**).

There are certainly limitations to emergent trends learned by masked pretraining. Unseen boundary conditions can be challenging; when pretrained on periodic 1D KdV-Burgers data, MAE models can only distinguish between periodic and non-periodic Heat equation samples without understanding differences between Dirichlet

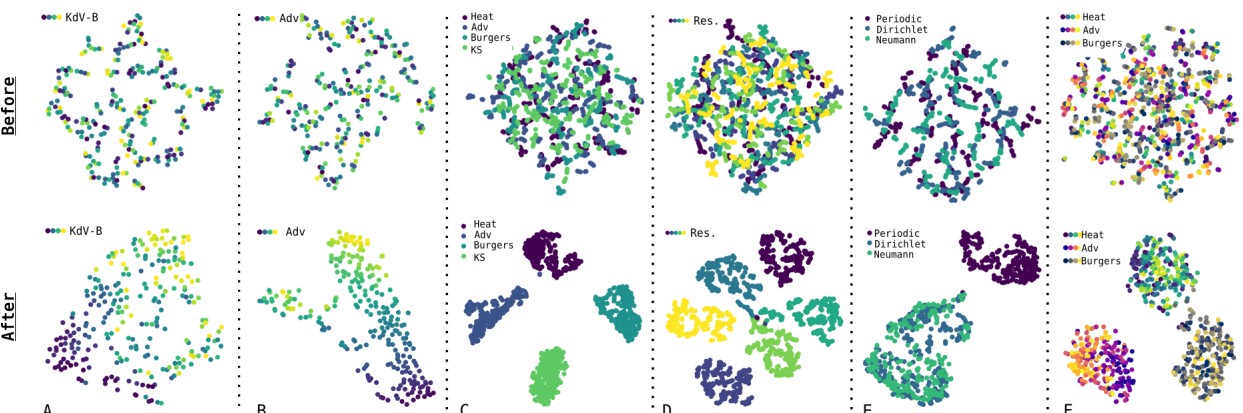

**Figure 4:** t-SNE embeddings of various PDEs. Plots show embeddings before and after using the MAE to encode samples, shown on the top and bottom. The MAE latent space shows structure despite not seeing labels of coefficients, PDEs, or BCs. **A:** 1D KdV-Burgers equation, colored by $\alpha$. **B:** 1D Advection equation, colored by $c$. **C:** 1D Heat, Burgers, Advection, and KS equations, colored by PDE. **D:** 1D KdV-Burgers equation, colored by resolution. **E:** 1D Heat equation, colored by boundary condition. **F:** 2D Heat, Advection and Burgers equations, colored by $\nu$ and $c$.

and Neumann BCs (Figure 4**E**). In addition, trends in 2D PDE data are more difficult to learn without labels. Despite separating between Heat, Adv, and Burgers samples, latent trends are only observed in 2D Advection samples based on wave speed **c** (Figure 4**F**).

### 5.3 PDE Feature Prediction

To evaluate the latent representations learned from masked pretraining, we regress PDE coefficients and classify PDE boundary conditions, equation families, and spatial resolutions from an initial time window. Regression tasks are separated by PDE (*KdV-B, KS, Heat, Adv, Burgers, NS*). Further, classification tasks are formulated as: ($Heat_{BC}$): predicting Periodic, Dirichlet, or Neumann BCs from the Heat equation, ($Wave_{BC}$): predicting Dirichlet or Neumann BCs from the Wave equation, ($PDEs$): predicting unseen PDEs from Heat, Adv, Burgers, and KS equation samples, and ($Res.$): predicting resolutions from $n_x = \{50, 60, 70, 80, 90, 100\}$ in 1D or $(n_x, n_y) = \{(48, 48), (52, 52), (56, 56), (60, 60), (64, 64)\}$ in 2D.

Several model variants are evaluated: a randomly initialized ViT baseline ($MAE_b$), a pretrained, frozen MAE encoder with a linear head ($MAE_f$), and a pretrained, fine-tuned MAE encoder (MAE). For these MAE models, regression and classification are performed using a CLS token and projecting the CLS embedding to the number of prediction features through a simple MLP. The baseline encoder is a ViT with the same model size and architecture as the pretrained MAE encoder, and when the MAE encoder is frozen, only the MLP head receives gradient updates. Additionally, we train MLP and CNN models to regress coefficients or classify PDE features in order to benchmark the difficulty of these tasks. Since these models cannot process variable sized inputs, the *Res.* experiment is not performed for these simpler baselines. Results are shown in Tables 3 & 4, with full error bars in Appendix E.1.

In 1D, the frozen $MAE_f$ encoder is able to outperform a supervised baseline on the *KdV-B*, *Heat*, *PDEs*, and *Res.* tasks despite never seeing labels throughout training. Further performance gains can be realized by allowing the MAE encoder to fine-tune on labeled data, and can outperform random initialization on unseen equations and BCs. An exception to this is the Advection and Wave equations. We hypothesize that these PDEs are heavily governed by the wave speed $c$ and boundary conditions (bouncing vs. reflecting waves) and are simple trends that supervised models can learn quickly. Indeed, the simple MLP and CNN baselines are able to achieve perfect classification accuracy on the Wave equation, likely since the wave either reflects or bounces off the boundaries according to the BC.

**Table 3:** 1D PDE feature prediction after MAE pretraining on the KdV-Burgers equation. Models are fine-tuned on 2000 held-out, labeled samples for each task. We consider regressing coefficients across four PDEs as well as classifying BCs of the heat/wave equation, identifying equations from a mixed set of PDEs, and sorting different spatial resolutions of a PDE. Regression errors are given as RMSE$\times 10^{-2}$ and classification errors are given as X-Ent$\times 10^{-4}$, averaged over 5 seeds.

| Model | KdV-B | Heat | Adv | KS | Heat$_{BC}$ | Wave$_{BC}$ | PDEs | Res. |
|---|---|---|---|---|---|---|---|---|
| MLP | 6.641 | 1.290 | 2.435 | 0.821 | 1.733 | **0.000** | 27.24 | — |
| CNN | 1.835 | 1.250 | 1.470 | 0.342 | 0.195 | **0.000** | 0.427 | — |
| MAE$_b$ | 3.454 | 0.834 | **0.241** | 0.354 | 0.123 | 0.022 | 0.355 | 64.52 |
| MAE$_f$ | 1.334 | 0.677 | 0.551 | 0.368 | 1.164 | 1.816 | 0.174 | 63.34 |
| MAE | **0.905** | **0.505** | 0.244 | **0.156** | **0.018** | 0.053 | **0.025** | **28.32** |

**Table 4:** 2D PDE feature prediction after MAE pretraining on a combined set of 2D Heat, Advection, and Burgers equations. Models are fine-tuned on 1024 held-out, labeled samples for each task, or 3072 samples in the combined case. Regression errors are given as RMSE$\times 10^{-1}$ and classification errors are given as X-Ent$\times 10^{-1}$, averaged over 5 seeds.

| Model | Heat | Adv | Burgers | Combined | NS | Res. |
|---|---|---|---|---|---|---|
| CNN | 0.305 | 1.057 | 1.370 | 0.371 | 1.224 | — |
| MAE$_b$ | 0.084 | **0.506** | 0.682 | 0.320 | 0.748 | 0.694 |
| MAE$_f$ | 0.232 | 0.540 | 0.606 | 0.384 | 0.709 | 0.636 |
| MAE | **0.062** | 0.507 | **0.409** | **0.265** | **0.594** | **0.005** |

In 2D, the effects of fine-tuning are more pronounced. Within the pretraining set, only in the 2D Burgers task does freezing the MAE encoder outperform a supervised baseline; nevertheless, masked pretraining serves as a good initialization for supervised fine-tuning. In addition, despite differing physics, prior knowledge from simpler 2D PDEs seems to benefit regression on the Navier-Stokes equations. When classifying 2D Heat, Advection, and Burgers data based on their discretization, MAE models greatly benefit from pretraining on multi-resolution data. We hypothesize that in 2D PDEs, variable spatial resolutions can be challenging to distinguish due to flattening the spatial dimension when patchifying inputs, whereas in 1D PDEs the data is already flattened.

### 5.4 Conditional Time-stepping and Super-resolution

**1D Experiments** To evaluate the use of the MAE encoder for practical tasks, we train a neural solver on various 1D PDEs to predict or upsample physics. For prediction, or time-stepping, models are given solutions are time $t$ and queried to predict solutions at at time $t + 1$. For upsampling, or super-resolution, models are given a low-resolution solution at time $t$ and queried to predict a high-fidelity solution at time $t$. For our experiments in 1D, we consider five PDEs (*KdV-B, Heat, Burgers, Adv, KS*) as well as two PDEs under varying boundary conditions (*Heat$_{BC}$, Wave$_{BC}$*) and predicting physics on various resolutions of the KdV-Burgers equation (*Res.*). Time-stepping is performed autoregressively by predicting multiple timesteps simultaneously to reduce error accumulation (Brandstetter et al., 2022). Furthermore, the pushforward trick (Brandstetter et al., 2022) is implemented. This adds model noise to inputs during training by making a prediction of a future timestep and using that prediction as a noised input the model; importantly gradients are not calculated for the initial pass. Lastly, we test on FNO (Li et al., 2020) and Unet architectures (Gupta and Brandstetter, 2022), (Ronneberger et al., 2015), and add conditioning information to hidden states after convolutions (Ho et al., 2020; Nichol and Dhariwal, 2021).

For super-resolution (SR), we implement a pipeline in which a network encodes low-resolution physics before upsampling with a discretization inversion operator $D^{-1}$ (linear interpolation in 1D and bicubic interpolation in 2D) and mapping to an output function space with a neural operator (Yang et al., 2023).Following this, we implement a Resnet encoder (Wang et al., 2021; Zhang et al., 2018) followed by an interpolation scheme and FNO operator; both Resnet and FNO models are provided conditioning information from MAE encodings of

**Table 5:** Conditional 1D PDE time-stepping and super-resolution. Models are trained on 2000 held-out fine-tuning samples to predict or upsample physics across several settings. Validation errors are reported as normalized L2 loss (time-stepping) or RMSE$\times 10^{-1}$ (SR) summed over all timesteps and averaged across five seeds.

| Model | KdV-B | Heat | Burgers | Adv | KS | Heat$_{BC}$ | Wave$_{BC}$ | Res. |
|---|---|---|---|---|---|---|---|---|
| FNO | 1.153 | 0.671 | 1.094 | 0.437 | 0.830 | 2.408 | 0.147 | 1.141 |
| FNO-MAE$_f$ | 1.043 | 0.655 | 1.121 | 0.431 | 0.821 | **1.747** | **0.135** | **1.018** |
| FNO-MAE | **1.037** | **0.643** | **0.952** | **0.294** | **0.812** | 1.846 | 0.148 | 1.07 |
| Unet | 0.823 | 0.420 | 0.649 | 0.194 | 1.333 | **4.249** | 0.747 | 0.766 |
| Unet-MAE$_f$ | 0.806 | 0.425 | 0.582 | **0.177** | 1.241 | 4.734 | 0.699 | 0.688 |
| Unet-MAE | **0.758** | **0.363** | **0.546** | 0.210 | **1.125** | 5.157 | **0.659** | **0.683** |
| Interp. | 0.540 | 0.345 | 1.357 | 0.231 | 3.599 | 0.225 | **0.347** | — |
| SR | 0.520 | 0.203 | 0.881 | 0.223 | 2.673 | 0.210 | 0.516 | — |
| SR-MAE$_f$ | 0.481 | 0.173 | 0.691 | **0.169** | 2.460 | 0.204 | 0.376 | — |
| SR-MAE | **0.475** | **0.151** | **0.676** | 0.194 | **2.422** | **0.170** | 0.349 | — |

low-resolution physics. The motivation behind using a two-step SR pipeline is to learn a vector embedding using the Resnet, then map from vector to function space with $T^{-1}$, and finally transform this function using an learnable operator to the output. For additional details on hyperparameters, see Appendix C. Additionally, we evaluate a simple baseline using linear interpolation to upsample low-resolution inputs (Interp.).

After pretraining an MAE encoder on the 1D KdV-Burgers equation, we compare the base neural solvers (FNO, Unet, SR) to conditioning on a frozen MAE embedding (-MAE$_f$) and allowing the MAE encoder to fine-tune when conditioning (-MAE). Results in 1D are presented in Table 5. Within the pretraining distribution (KdV-B) and certain PDEs, MAE conditioning consistently improves time-stepping and super-resolution performance. In addition, allowing MAE encoders to fine-tune can further lower errors. However, there are various exceptions, in particular PDEs with unseen boundary conditions. Despite this, improvements are consistent across different neural solver architectures, suggesting that pretrained MAE models can be agnostic to downstream model choices. In addition, in 1D, SR results are less significant suggesting that simple interpolation schemes are often enough for these phenomena, especially for simple equations such as the advection or wave PDEs.

**2D Experiments** Following the setup in 1D, we repeat time-stepping/super-resolution experiments on 2D PDEs (*Heat, Adv, Burgers, NS*) and a combined set of 2D Heat, Advection, and Burgers equations (*Combined*). Additionally, we evaluate time-stepping performance on the combined 2D Heat, Advection, and Burgers equations discretized at variable resolutions (*Res.*). We follow the same conditioning and training strategies as 1D experiments, but modify the architectures to support 2D inputs, and present results in Table 6. Additionally, the interpolation baseline implements bicubic upsampling (Interp.). After pretraining an MAE encoder on the 2D Heat, Advection, and Burgers equations, we observe improvements in conditional physics prediction and upsampling. Improvements tend to be more pronounced in 2D; we hypothesize that the increased difficulty of the task increases the importance of MAE encoder guidance in time-stepping and super-resolution. However, out-of-distribution datasets are still challenging: when extrapolating pretrained encoders to new PDEs, such as the Navier-Stokes equations, the performance is limited. Nevertheless, we observe similar trends whereby MAE conditioning is agnostic to downstream architectures.

**Additional Benchmarks** We consider two additional benchmarks: a randomly initialized ViT encoder that embeds PDE inputs to a conditioning vector as well as a linear model that encodes ground-truth coefficient or boundary condition information to a conditioning vector. We present detailed results in Appendix E.2 and E.4, and discuss the overall results here. In general, we observe that the randomly initialized, fine-tuned encoder also improves PDE prediction and upsampling, and this improvement generally matches or outperforms the performance of the frozen MAE encoder. However, allowing the MAE encoder to fine-tune generally outperforms this random initialization and approaches the linear benchmark.

**Table 6:** Conditional 2D PDE time-stepping and super-resolution. Models are trained on 1024 held-out fine-tuning samples, or 3072 in the combined case, to predict or upsample physics. Validation errors are reported as normalized L2 loss (time-stepping) or RMSE$\times 10^{-1}$ (SR) summed over all timesteps and averaged across three seeds.

| Model | Heat | Adv | Burgers | Combined | NS | Res. |
|---|---|---|---|---|---|---|
| FNO | 0.427 | 2.301 | 0.417 | 0.978 | **0.466** | 1.006 |
| FNO-MAE$_f$ | 0.233 | 1.179 | 0.252 | 0.607 | 0.59 | 0.701 |
| FNO-MAE | **0.128** | **1.135** | **0.198** | **0.494** | 0.477 | **0.499** |
| Unet | 0.147 | 1.795 | 0.226 | 0.835 | 0.713 | 0.908 |
| Unet-MAE$_f$ | 0.136 | 1.804 | 0.229 | 0.761 | **0.669** | 0.861 |
| Unet-MAE | **0.116** | **1.230** | **0.186** | **0.669** | 0.692 | **0.676** |
| Interp. | 0.492 | 0.937 | 0.488 | 0.673 | 0.367 | — |
| SR | 0.175 | 2.014 | 0.295 | 0.534 | **0.326** | — |
| SR-MAE$_f$ | 0.159 | 0.659 | 0.264 | **0.407** | 0.347 | — |
| SR-MAE | **0.152** | **0.639** | **0.253** | 0.472 | 0.337 | — |

Although the linear benchmark generally performs the best in 1D, there are certain exceptions to this. Equations dominated by the coefficient response generally suffer from coefficient conditioning; we observe that the model heavily overfits to the true coefficient and does not learn the underlying PDE dynamics. Furthermore, for PDEs that do not have coefficient information, such as the 1D inviscid Burgers equation, this linear benchmark cannot be applied. In these cases, MAE encodings can still improve performance, suggesting that there are latent PDE features beyond coefficient information that neural solvers can benefit from. Lastly, in 2D the linear benchmark performs much worse, only achieving the lowest error in a few scenarios, and in some cases harming performance of the base model. This could be because PDE dynamics becomes much more complex in 2D and relies less on coefficient information, which is a low-dimensional vector and provides sparse information.

Lastly, to benchmark our method against other pretraining methods, we consider pretraining an encoder using a contrastive self-supervised technique proposed by Mialon et al. (2023), which relies on using Lie symmetries to cluster PDE data in a learned latent space. We contrastively pretrain an encoder on 1D KdV-Burgers data and the evaluate conditional timestepping performance on various downstream 1D PDEs. We present these results in Appendix E.3. To summarize, our approach is on par with Lie contrastive pretraining for PDE samples within the pretraining distribution; however, when extrapolating to unseen PDEs, masked pretraining is able to outperform contrastive methods.

## 6  Discussion

In this section, we discuss results and provide additional insight. Although masking as a pretraining strategy is not physically valid, this requirement does not seem to be necessary for learning. Both in this study, and in related works (Hao et al., 2024; Zhou et al., 2024; Rahman et al., 2024), noising or masking strategies are used to improve model performance, which both represent artificial phenomena. Within the context of masking, it is also natural to ask if a unique solution exists given a masked input. Certainly at the extreme where all of the input is masked the solution is not unique; however, it seems empirically that only a small amount of information (25% in 1D and 10% in 2D) is needed, which is corroborated by related work in the CV domain (Feichtenhofer et al.). We can observe this through the validation error: if this quantity is small, a unique solution is being regressed in the validation set.

While promising, the presented results in time-stepping and super-resolution would likely not outperform a foundation model trained specifically for these tasks (Hao et al., 2024; Herde et al., 2024). This is because direct transfer learning has been shown to outperform surrogate objectives when pretraining for PDEs (Zhou et al., 2024); one hypothesized reason for this is the lack of abundant unlabeled data in the PDE domain (or equivalently, the downstream task uses also unlabeled data). However, encoder-style approaches such

as this work or contrastive PDE encoders (Mialon et al., 2023; Zhang et al., 2023) are more flexible than foundation models, capable of being applied to arbitrary downstream architectures and different fine-tuning tasks. Additionally, when using a surrogate objective to train an encoder, models can learn more general latent representations, compared to using a neural solver or foundation model to only predict the next step of a PDE rollout. We show some preliminary results demonstrating this in Appendix F. While this versatility is interesting, the practicality of this is unclear, since time-stepping is so singularly important. However, this research direction is still underexplored and perhaps future work will find an interesting set of uses for PDE encoders.

Within the context of masked pretraining, there are a few additional limitations to be recognized. It can be costly to fully fine-tune the pretrained encoder during time-stepping or super-resolution since it operates on all unmasked tokens, and as a result does not benefit from the speed gains during MAE pretraining. This makes full fine-tuning likely infeasible when compared to baselines. To address this, freezing the MAE encoder greatly improves the training speed but decreases performance, especially on unseen PDEs. With enough PDE data during pretraining, freezing MAE encoders can be more practical since the pretraining distribution would cover most downstream cases. Lastly, for simpler 1D dynamics when coefficient information is available, conditioning on ground-truth PDE parameters remains the best choice in most scenarios. However, these approaches are not exclusive; initial work suggests that models provided with both ground-truth information and MAE embeddings can outperform models provided with just one of either.

## 7 Conclusion

We have presented a new method that extends masked pretraining from vision and video domains to physics, and evaluated several PDE-specific modifications. We empirically validate MAE models through reconstructing a diverse set of 1D and 2D PDEs and show limited generalization behavior to different spatial discretizations and unseen equations. Furthermore, we evaluate the latent representations learned during MAE training and find structured trends that can be used to predict PDE features. In practice, MAE encoders can also be used to improve time-stepping and super-resolution tasks across diverse physics scenarios.

A promising direction would be to scale MAE models to larger datasets, as the current approach exhibits the same scalability as the originally proposed MAE (He et al., 2021). Additionally, future work could explore masked modeling approaches in more complex 2D and 3D problems. Lastly, future work could explore manipulating latent physics to generate new solutions or performing arithmetic in an autoencoder latent space. We present a potential setup for this in Appendix D which relies on encoding solutions from separate equations, adding them in latent space, and decoding this latent vector.

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

**Table 7:** MAE Hyperparameters during pretraining.

**(a)** 1D PDEs

| Parameters | Value |
|---|---|
| Batch Size | 256 |
| Epochs | 20 |
| Encoder Dim | 256 |
| Decoder Dim | 32 |
| Patch Size | (5, 5) |
| Masking Ratio | 0.75 |
| Time Window | 20 |
| Augmentation Ratio | 0.5 |
| Base LR | 1e-3 |
| Optimizer | AdamW |
| Scheduler | OneCycleLR |

**(b)** 2D PDEs

| Parameters | Value |
|---|---|
| Batch Size | 64 |
| Epochs | 20 |
| Encoder Dim | 256 |
| Decoder Dim | 32 |
| Patch Size | (4, 4, 4) |
| Masking Ratio | 0.90 |
| Time Window | 16 |
| Augmentation Ratio | 0.5 |
| Base LR | 1e-3 |
| Optimizer | AdamW |
| Scheduler | OneCycleLR |

## A  MAE Implementation

To implement the MAE encoder and decoder, we use a ViT architecture (47), which uses a self-attention layer (20) and MLP, both with LayerNorms (74). We present hyperparameters in Table 7. To study the effects of various hyperparameters, including model size, masking ratio, and patch size, we run ablation studies on masked reconstruction of 1D PDEs, and report reconstruction MSE errors on a validation set in Table 8. Overall, we find that increasing model size in limited data regimes—only 10000 KdV-Burgers samples were used in pretraining—tends to contribute to overfitting and increases validation errors. Predictably, increasing the masking ratio increases reconstruction errors as a result of less information being provided to MAE models. Furthermore, decreasing the patch size reduces errors but requires a higher computational cost, which is consistent with results in CV domains (56).

In 1D, the MAE is trained on a single NVIDIA GeForce RTX 2080 Ti, and reaches convergence in about 6 hours. In 2D, the MAE is trained on a single NVIDIA RTX A6000, and reaches convergence in about 24 hours. Model size, masking ratio, and patch size all affect the computational cost and can be used to tradeoff performance for compute and memory.

To motivate the ViT architecture, we investigate the effect of different model choices and hyperparameters on performance. We find that FNO autoencoders tend to have poor reconstruction capabilities due to introducing spurious high-frequency modes when masking spatially. Unet approaches fare better but still suffer sharp boundaries across masked and unmasked regions. Additionally, we evaluate different ViT variants, such as ViViT (axial attention) and Swin Transformer (window attention). We find that restricting the attention mechanism reduces performance, and the additional speedups were not significant since masking out large portions of the input already reduces computation. Lastly, model performance tends to vary smoothly with changes in hyperparameters; for example, reducing patch size slightly increases performance across downstream tasks.

**Table 8:** MAE model ablation studies on the 1D KdV-Burgers equation.

**(a)** Model Size

| # Params | Error |
|---|---|
| 1M | 2.37e-03 |
| 5M | 2.48e-03 |
| 25M | 3.36e-03 |

**(b)** Masking Ratio

| Masking Ratio | Error |
|---|---|
| 0.6 | 8.02e-04 |
| 0.75 | 1.66e-03 |
| 0.90 | 6.68e-03 |

**(c)** Patch Size, in $(p_t, p_x)$

| Patch Size | Error |
|---|---|
| (5, 5) | 1.12e-03 |
| (4, 4) | 7.05e-04 |
| (4, 2) | 6.46e-04 |
| (2, 4) | 4.79e-04 |

## B  Additional MAE Examples

### B.1  Additional 1D MAE Predictions

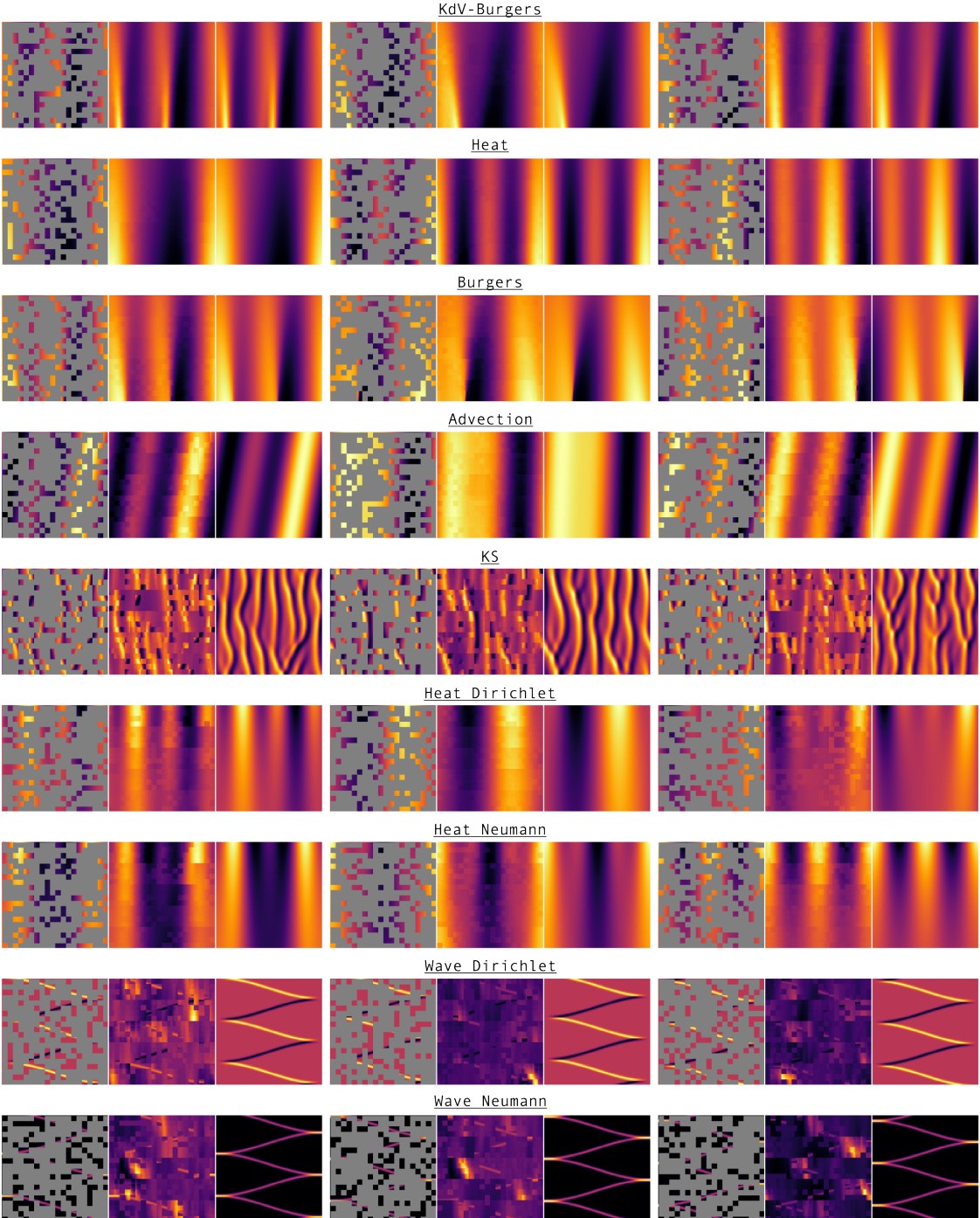

**Figure 5:** Additional 1D MAE Reconstruction examples after pretraining on the 1D KdV-Burgers equation. Each triplet is shown with the masked sample (Left), MAE reconstruction (Middle), and ground truth PDE (right). We include additional reconstructions of unseen boundary conditions for the Heat and Wave equations.

## B.2 Additional 2D MAE Predictions

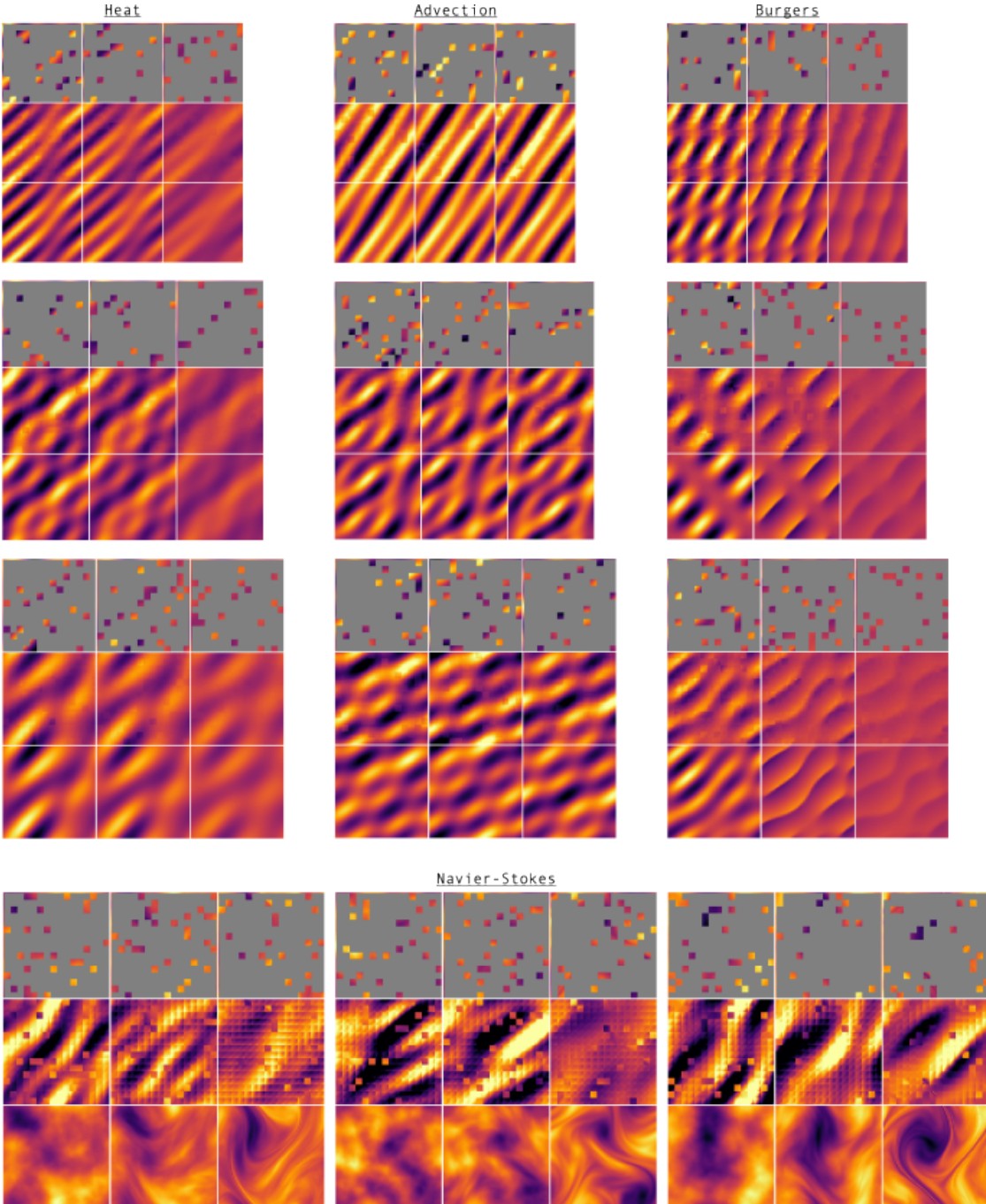

**Figure 6:** Additional 2D MAE Reconstruction examples after pretraining on the 2D Heat, Advection, and Burgers Equations. Each sample is shown with the masked sample (Top), MAE reconstruction (Middle), and ground truth PDE (Bottom). We include sample MAE predictions at variable resolutions for the 2D Heat, Advection, and Burgers equations; the lowest resolution (top) is $(48, 48)$, the medium resolution (middle) is $(52, 52)$, and the high resolution (bottom) is $(56, 56)$ We include additional reconstructions of the incompressible NS equations at the native resolution $(64, 64)$.

### B.3 2D Smoke Buoyancy Predictions

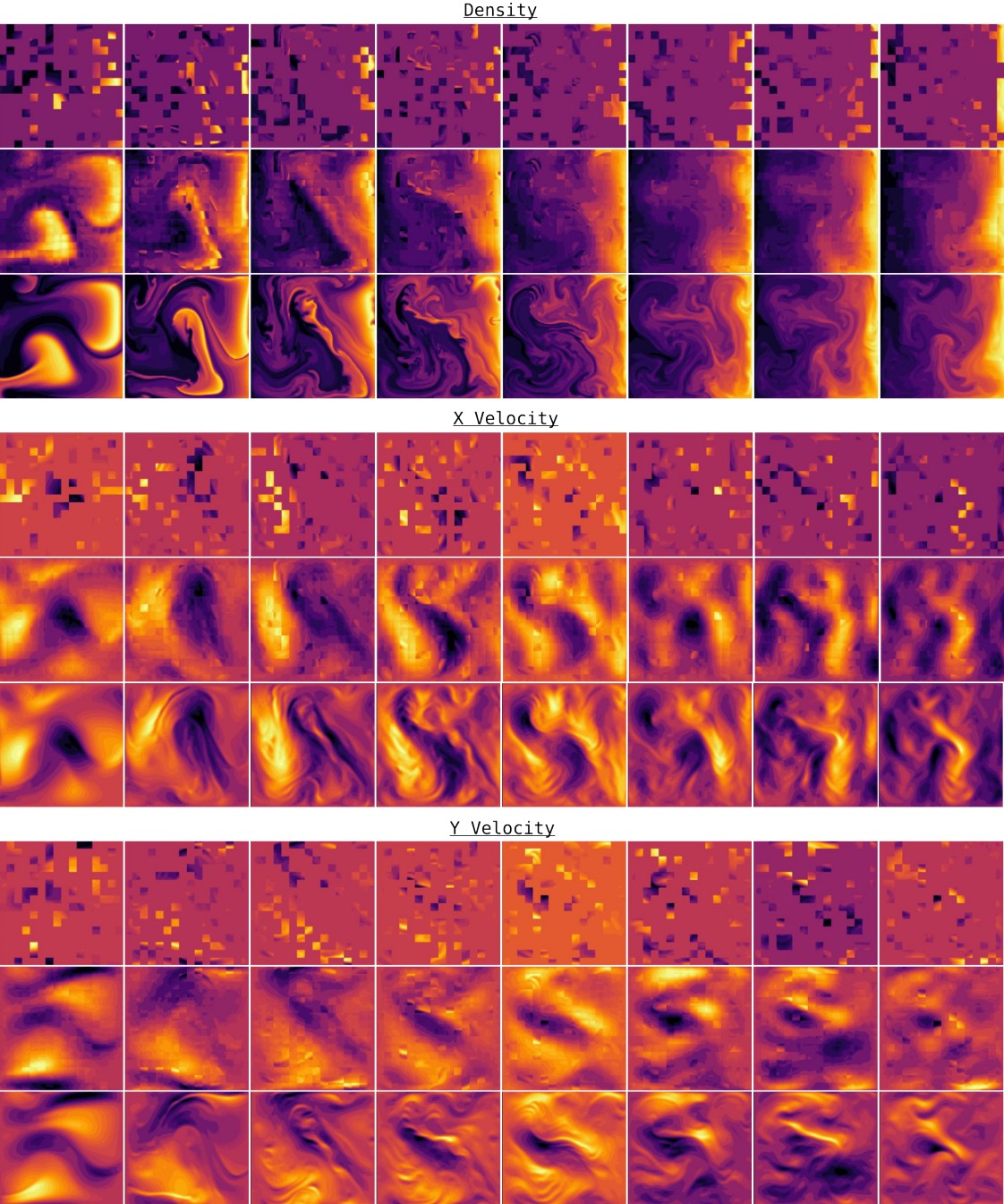

**Figure 7:** MAE validation reconstructions after training on 2D Navier-Stokes data with variable buoyancy factors (23). The MAE model is trained on a resolution of $(n_t, n_x, n_y) = (56, 128, 128)$ with three data channels $(\rho, v_x, v_y)$ and a masking ratio of 0.75. Triplets are shown with the masked input (top), MAE reconstruction (middle), and ground truth (bottom), with the top, middle, and bottom triplets displaying density, X velocity, and Y velocity. The complex dynamics is challenging; indeed, many of the fine details are lost in the MAE reconstruction. We train with a larger model (45M params) and patch size $(2, 8, 8)$, which takes around 9 hours on a NVIDIA RTX A6000 GPU.

**Table 9:** Hyperparameters for architectures used for time-stepping and super-resolution.

**(a)** FNO Hyperparameters

| Parameter | 1D/2D |
|---|---|
| Modes | 24/12 |
| Width | 64/48 |
| # Layers | 4 |
| Cond. dim | 32 |
| Conditioning | Add |
| Init lr | 8e-4 |
| # Params | 1M/5M |

**(b)** Unet Hyperparameters

| Parameter | 1D/2D |
|---|---|
| Hidden channels | 16 |
| Channel mults. | (1, 2, 4) |
| Cond. dim | 32 |
| Conditioning | AdaGN |
| Init lr | 8e-4 |
| # Params | 1M/2M |

**(c)** Resnet Hyperparameters

| Parameter | 1D/2D |
|---|---|
| Hidden channels | 64 |
| # Blocks | 4 |
| Cond. dim | 32 |
| Conditioning | Add |
| Init lr | 8e-4 |
| # Params | 1M/3M |

## C    Training Details

Hyperparameters used for the FNO and Unet models during time-stepping are presented in Table 9. Additionally, for the SR pipeline, hyperparameters used for the Resnet encoder, which uses residual dense blocks (71), and FNO operator are reported in Table 9. We present a schematic of the SR pipeline in Figure 8.

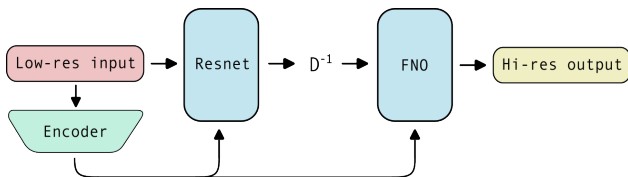

**Figure 8:** Conditional super-resolution pipeline.

## D    Latent Arithmetic

MAE encoders have shown strong capabilities in extracting information from self-supervised PDE learning, creating opportunities to operate in this latent space. This could be beneficial since many PDEs are compositions of simpler phenomena, and recombining PDE solutions in latent space may result in obtaining novel PDE solutions for free. After pretraining on the 1D KdV-Burgers equation, we consider an arithmetic task where samples of the Heat and Burgers equation are embedded and added in the latent space before being decoded to a novel solution (Figure 9). Concretely, we generate 1D Heat data from the PDE: $\partial_t u - \nu \partial_{xx} u = 0$, 1D inviscid Burgers data from the PDE: $\partial_t u + u \partial_x u = 0$, and 1D viscous Burgers data from adding the two PDEs: $\partial_t u - \nu \partial_{xx} u + u \partial_x u = 0$. When given identical initial conditions and coefficients, PDE reconstructions obtained from summing latent Heat and Burgers embeddings qualitatively resemble solutions of the viscous Burgers PDE. In addition, if different latent embeddings can be added, the weighting of each embedding can be varied to control the resemblance of the reconstruction to make interpolated samples that have more shock formation or diffusive behavior (e.g., more resemble the Burgers or Heat equation).

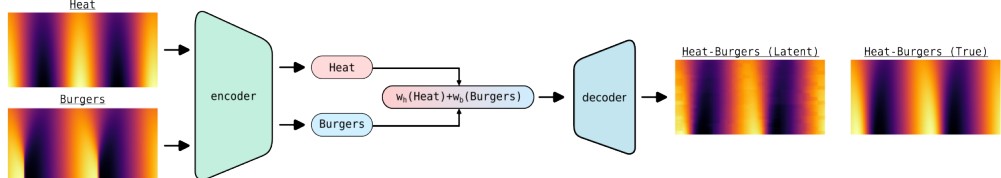

**Figure 9:** A proposed setup for operating on latent PDE vectors. PDE data is encoded after masked pretraining and summed in the latent space before being decoded. These reconstructions can approximate summed PDEs in physical space.

# E  Additional Results and Statistical Significance

## E.1  Feature Prediction

**Table 10:** 1D PDE coefficient regression. Validation errors are given as RMSE $\times 10^{-2}$. The mean error and standard deviation are calculated over five seeds; error bars are reported as one standard deviation above or below the mean. Lowest errors that are statistically significant are bolded.

| Model | KdV-B | Heat | Adv | KS |
|---|---|---|---|---|
| $\text{MAE}_\text{b}$ | $3.454 \pm 0.131$ | $0.834 \pm 0.041$ | $0.241 \pm 0.051$ | $0.354 \pm 0.104$ |
| $\text{MAE}_\text{f}$ | $1.334 \pm 0.036$ | $0.677 \pm 0.016$ | $0.551 \pm 0.03$ | $0.368 \pm 0.02$ |
| MAE | $\mathbf{0.905 \pm 0.059}$ | $\mathbf{0.505 \pm 0.065}$ | $0.244 \pm 0.064$ | $\mathbf{0.156 \pm 0.023}$ |

**Table 11:** 1D PDE feature classification. Validation Errors are given as X-Ent $\times 10^{-4}$. The mean error and standard deviation are calculated over five seeds; error bars are reported as one standard deviation above or below the mean. Lowest errors that are statistically significant are bolded.

| Model | $\text{Wave}_\text{BC}$ | $\text{Heat}_\text{BC}$ | PDEs | Res. |
|---|---|---|---|---|
| $\text{MAE}_\text{b}$ | $0.022 \pm 0.004$ | $0.123 \pm 0.108$ | $0.355 \pm 0.276$ | $64.52 \pm 3.475$ |
| $\text{MAE}_\text{f}$ | $1.817 \pm 1.175$ | $1.164 \pm 0.838$ | $0.174 \pm 0.064$ | $63.34 \pm 3.71$ |
| MAE | $0.053 \pm 0.048$ | $0.018 \pm 0.002$ | $\mathbf{0.025 \pm 0.013}$ | $\mathbf{28.322 \pm 23.351}$ |

Detailed results for 1D PDE feature prediction tasks are reported in Tables 10 and 11. For 1D tasks, certain experiments have high variance; we hypothesize that this is due to the fact that each seed samples a random dataset of 2000 samples from a much larger dataset. This would make some seeds easier to regress/classify than others, but within each seed the models follow trends consistent with the mean statistics. Furthermore, the magnitude of the X-Ent error is very small, leading to high variations after the model has learned most of the relevant features.

**Table 12:** 2D PDE coefficient regression and feature classification. Validation errors are reported as RMSE $\times 10^{-1}$ for regression tasks and X-Ent $\times 10^{-1}$ for classification tasks. The mean error and standard deviation are calculated over five seeds; error bars are reported as one standard deviation above or below the mean. Lowest errors that are statistically significant are bolded.

| Model | Heat | Adv | Burgers |
|---|---|---|---|
| $\text{MAE}_\text{b}$ | $0.084 \pm 0.014$ | $0.506 \pm 0.009$ | $0.682 \pm 0.037$ |
| $\text{MAE}_\text{f}$ | $0.232 \pm 0.01$ | $0.54 \pm 0.015$ | $0.606 \pm 0.012$ |
| MAE | $\mathbf{0.062 \pm 0.003}$ | $0.507 \pm 0.006$ | $\mathbf{0.409 \pm 0.008}$ |
| Model | Combined | NS | Res. |
| $\text{MAE}_\text{b}$ | $0.320 \pm 0.007$ | $0.748 \pm 0.005$ | $0.694 \pm 0.174$ |
| $\text{MAE}_\text{f}$ | $0.384 \pm 0.009$ | $0.709 \pm 0.01$ | $0.636 \pm 0.061$ |
| MAE | $\mathbf{0.265 \pm 0.007}$ | $\mathbf{0.594 \pm 0.038}$ | $\mathbf{0.005 \pm 0.002}$ |

Detailed results for 2D feature prediction tasks are reported in Table 12. The 2D results tend to be more consistent and have lower variance, since a fixed dataset was used for each seed and only the shuffling is changed.

## E.2 Time-stepping

**Table 13:** Conditional 1D PDE time-stepping. Validation errors are reported as normalized L2 loss summed over all PDE timesteps. The mean error and standard deviation are calculated over five seeds; error bars are reported as one standard deviation above or below the mean. Lowest errors that are statistically significant are bolded.

| Model | KdV-B | Heat | Burgers | Adv |
|---|---|---|---|---|
| FNO | $1.132 \pm 0.037$ | $0.671 \pm 0.039$ | $1.094 \pm 0.060$ | $0.437 \pm 0.052$ |
| FNO-Enc | $1.041 \pm 0.029$ | $0.644 \pm 0.038$ | $1.129 \pm 0.062$ | $0.347 \pm 0.074$ |
| FNO-MAE$_f$ | $1.077 \pm 0.060$ | $0.655 \pm 0.008$ | $1.121 \pm 0.051$ | $0.431 \pm 0.054$ |
| FNO-MAE | $1.060 \pm 0.032$ | $0.643 \pm 0.035$ | $\mathbf{0.952 \pm 0.038}$ | $0.294 \pm 0.033$ |
| FNO-Lin | $\mathbf{0.936 \pm 0.029}$ | $0.75 \pm 0.062$ | N/A | $\mathbf{0.204 \pm 0.019}$ |
| Unet | $0.872 \pm 0.069$ | $0.420 \pm 0.021$ | $0.649 \pm 0.052$ | $0.194 \pm 0.059$ |
| Unet-Enc | $0.834 \pm 0.043$ | $0.395 \pm 0.021$ | $0.582 \pm 0.032$ | $0.224 \pm 0.057$ |
| Unet-MAE$_f$ | $0.833 \pm 0.038$ | $0.425 \pm 0.012$ | $0.582 \pm 0.017$ | $0.177 \pm 0.012$ |
| Unet-MAE | $0.795 \pm 0.040$ | $\mathbf{0.363 \pm 0.010}$ | $0.546 \pm 0.024$ | $0.21 \pm 0.047$ |
| Unet-Lin | $\mathbf{0.659 \pm 0.045}$ | $0.445 \pm 0.008$ | N/A | $0.166 \pm 0.032$ |

| Model | KS | Heat$_{BC}$ | Wave$_{BC}$ | Res. |
|---|---|---|---|---|
| FNO | $0.83 \pm 0.028$ | $0.147 \pm 0.015$ | $2.408 \pm 0.848$ | $1.141 \pm 0.021$ |
| FNO-Enc | $0.82 \pm 0.082$ | $0.133 \pm 0.019$ | $2.012 \pm 1.194$ | $1.038 \pm 0.037$ |
| FNO-MAE$_f$ | $0.821 \pm 0.088$ | $0.135 \pm 0.013$ | $1.747 \pm 0.665$ | $1.018 \pm 0.13$ |
| FNO-MAE | $0.812 \pm 0.061$ | $0.148 \pm 0.015$ | $1.846 \pm 1.885$ | $1.070 \pm 0.011$ |
| FNO-Lin | $0.757 \pm 0.077$ | $0.132 \pm 0.020$ | $1.454 \pm 0.450$ | $\mathbf{0.899 \pm 0.01}$ |
| Unet | $1.333 \pm 0.068$ | $0.747 \pm 0.043$ | $4.249 \pm 2.296$ | $0.766 \pm 0.083$ |
| Unet-Enc | $1.203 \pm 0.102$ | $0.691 \pm 0.046$ | $4.902 \pm 1.935$ | $0.739 \pm 0.088$ |
| Unet-MAE$_f$ | $1.241 \pm 0.055$ | $0.699 \pm 0.023$ | $4.734 \pm 2.135$ | $0.688 \pm 0.077$ |
| Unet-MAE | $1.125 \pm 0.029$ | $0.659 \pm 0.047$ | $5.157 \pm 1.760$ | $0.683 \pm 0.087$ |
| Unet-Lin | $1.172 \pm 0.039$ | $0.717 \pm 0.027$ | $4.727 \pm 2.093$ | $0.573 \pm 0.095$ |

Following the main paper, we introduce two conditional benchmarks. We evaluate a randomly initialized and fine-tuned ViT encoder with the same architecture as the MAE encoder (-Enc), as well as a linear encoder that embeds the ground-truth PDE parameters as the conditioning information (-Lin).

In 1D, time-stepping results tend to have high variance; however, overall trends are still consistent with those reported in the main body. The variance is likely attributed to variations in the dataset for each seed; each seed samples a different set of 2000 samples from a larger PDE dataset, and as a result, some data splits may be easier than others. This results in the variance being high across seeds, however, within a seed (i.e. within a dataset), model performance closely follows trends consistent with the mean statistics.

**Table 14:** Conditional 2D PDE time-stepping. Validation errors are reported as normalized L2 loss summed over all PDE timesteps. The mean error and standard deviation are calculated over three seeds; error bars are reported as one standard deviation above or below the mean. Lowest errors that are statistically significant are bolded.

| Model | Heat | Adv | Burgers |
|---|---|---|---|
| FNO | $0.427 \pm 0.006$ | $2.301 \pm 0.094$ | $0.417 \pm 0.063$ |
| FNO-Enc | $0.152 \pm 0.013$ | $1.909 \pm 0.399$ | $0.241 \pm 0.032$ |
| FNO-MAE$_f$ | $0.233 \pm 0.028$ | $1.179 \pm 0.036$ | $0.252 \pm 0.012$ |
| FNO-MAE | $0.128 \pm 0.008$ | $1.135 \pm 0.121$ | $0.198 \pm 0.009$ |
| FNO-Lin | $0.118 \pm 0.005$ | $2.531 \pm 0.013$ | $\mathbf{0.149 \pm 0.036}$ |
| Unet | $0.147 \pm 0.031$ | $1.795 \pm 0.105$ | $0.226 \pm 0.018$ |
| Unet-Enc | $0.132 \pm 0.040$ | $1.604 \pm 0.164$ | $0.218 \pm 0.02$ |
| Unet-MAE$_f$ | $0.136 \pm 0.009$ | $1.804 \pm 0.066$ | $0.229 \pm 0.017$ |
| Unet-MAE | $0.116 \pm 0.031$ | $\mathbf{1.23 \pm 0.161}$ | $\mathbf{0.186 \pm 0.011}$ |
| Unet-Lin | $0.153 \pm 0.043$ | $2.571 \pm 0.011$ | $0.215 \pm 0.006$ |
| Model | Combined | NS | Res. |
| FNO | $0.978 \pm 0.055$ | $0.466 \pm 0.014$ | $1.006 \pm 0.02$ |
| FNO-Enc | $0.767 \pm 0.028$ | $0.514 \pm 0.123$ | $0.709 \pm 0.055$ |
| FNO-MAE$_f$ | $0.607 \pm 0.019$ | $0.59 \pm 0.107$ | $0.701 \pm 0.051$ |
| FNO-MAE | $\mathbf{0.494 \pm 0.043}$ | $0.477 \pm 0.029$ | $\mathbf{0.499 \pm 0.024}$ |
| FNO-Lin | $0.977 \pm 0.021$ | $0.445 \pm 0.026$ | $0.986 \pm 0.015$ |
| Unet | $0.835 \pm 0.067$ | $0.713 \pm 0.005$ | $0.908 \pm 0.061$ |
| Unet-Enc | $0.791 \pm 0.061$ | $0.695 \pm 0.027$ | $0.971 \pm 0.023$ |
| Unet-MAE$_f$ | $0.761 \pm 0.051$ | $0.669 \pm 0.031$ | $0.861 \pm 0.028$ |
| Unet-MAE | $\mathbf{0.669 \pm 0.015}$ | $0.692 \pm 0.039$ | $\mathbf{0.676 \pm 0.064}$ |
| Unet-Lin | $1.013 \pm 0.03$ | $\mathbf{0.635 \pm 0.002}$ | $1.098 \pm 0.026$ |

In 2D, time-stepping results have much lower variance; this is likely due to the fact that each seed uses the same dataset, with only the shuffling changing. Furthermore, the linear benchmark is less effective; in most experiments a learned encoding can outperform ground-truth PDE parameters, especially when predicting a combined or multi-resolution dataset of PDEs.

### E.3 Comparison to Contrastive Learning with Lie Augmentations

**Table 15:** We compare our approach to a contrastive self-supervised approach. After training a masked and contrastive encoder on the KdV-B pretraining set, we compare conditioning an FNO backbone to time-step different downstream 1D PDEs. The MAE encoder shows comparable performance within the pretraining set (KdV-B), but has better generalization behavior to unseen PDEs. Validation errors are reported as normalized L2 loss summed over all PDE timesteps

| Model | KdV-B | Heat | Burgers | Adv |
|---|---|---|---|---|
| FNO | 1.506 | 0.827 | 1.386 | 0.567 |
| FNO-Contrastive | **1.171** | 0.918 | 0.916 | 0.555 |
| FNO-MAE | 1.183 | **0.721** | **0.831** | **0.299** |

### E.4 Super-resolution

**Table 16:** Conditional 1D super-resolution. Validation errors are reported as RMSE $\times 10^{-1}$ summed over all PDE timesteps. The mean error and standard deviation are calculated over five seeds; error bars are reported as one standard deviation above or below the mean. Lowest errors that are statistically significant are bolded.

| Model | KdV-B | Heat | Burgers | Adv |
|---|---|---|---|---|
| SR | $0.520 \pm 0.021$ | $0.203 \pm 0.011$ | $0.881 \pm 0.062$ | $0.223 \pm 0.004$ |
| SR-Enc | $0.489 \pm 0.022$ | $0.166 \pm 0.021$ | $0.642 \pm 0.074$ | $0.202 \pm 0.003$ |
| SR-MAE$_f$ | $0.481 \pm 0.039$ | $0.173 \pm 0.015$ | $0.691 \pm 0.027$ | $0.169 \pm 0.032$ |
| SR-MAE | $0.475 \pm 0.018$ | $0.151 \pm 0.028$ | $0.676 \pm 0.060$ | $0.194 \pm 0.016$ |
| SR-Lin | $0.484 \pm 0.017$ | $0.131 \pm 0.017$ | N/A | $\mathbf{0.133 \pm 0.013}$ |

| Model | KS | Heat$_{BC}$ | Wave$_{BC}$ |
|---|---|---|---|
| SR | $2.673 \pm 0.101$ | $0.210 \pm 0.016$ | $0.516 \pm 0.015$ |
| SR-Enc | $2.585 \pm 0.062$ | $0.174 \pm 0.01$ | $0.373 \pm 0.022$ |
| SR-MAE$_f$ | $2.460 \pm 0.056$ | $0.204 \pm 0.015$ | $0.376 \pm 0.032$ |
| SR-MAE | $2.422 \pm 0.052$ | $0.170 \pm 0.019$ | $0.349 \pm 0.022$ |
| SR-Lin | $2.517 \pm 0.038$ | $0.177 \pm 0.027$ | $0.451 \pm 0.014$ |

Differences between benchmarks for 1D super-resolution tend to be incremental. Despite this, using a frozen MAE encoding remains a simple method to improve performance with a negligible training cost. In general, super-resolution for 1D PDEs is a relatively easy task, and changes in model architecture do not significantly affect results.

**Table 17:** Conditional 2D super-resolution. Validation errors are reported as RMSE $\times 10^{-1}$ summed over all PDE timesteps. he mean error and standard deviation are calculated over three seeds; error bars are reported as one standard deviation above or below the mean. Lowest errors that are statistically significant are bolded.

| Model | Heat | Adv | Burgers |
|---|---|---|---|
| SR | $0.175 \pm 0.023$ | $2.014 \pm 0.205$ | $0.295 \pm 0.018$ |
| SR-Enc | $0.152 \pm 0.01$ | $0.804 \pm 0.052$ | $0.252 \pm 0.047$ |
| SR-MAE$_f$ | $0.159 \pm 0.007$ | $0.659 \pm 0.089$ | $0.264 \pm 0.05$ |
| SR-MAE | $0.152 \pm 0.004$ | $0.639 \pm 0.125$ | $0.253 \pm 0.017$ |
| SR-Lin | $0.167 \pm 0.015$ | $2.016 \pm 0.015$ | $0.263 \pm 0.021$ |

| Model | Combined | NS |
|---|---|---|
| SR | $0.534 \pm 0.037$ | $0.326 \pm 0.039$ |
| SR-Enc | $0.49 \pm 0.016$ | $0.364 \pm 0.006$ |
| SR-MAE$_f$ | $0.407 \pm 0.002$ | $0.347 \pm 0.007$ |
| SR-MAE | $0.472 \pm 0.005$ | $0.337 \pm 0.012$ |
| SR-Lin | $1.235 \pm 0.032$ | $0.366 \pm 0.003$ |

In 2D, the general trend remains similar to 1D results; clear model choices for super-resolution are not apparent. Despite this, using a frozen MAE encoding often outperforms the linear benchmark; this can be a good way to boost model performance without additional training cost when PDE samples are within the pretraining distribution.

# F  Comparison of Latent Embeddings

To understand the versatility of masking pretraining, we would like to consider what it means to be a PDE learner beyond predicting future physics. For example, when reasoning about physics, humans can add, subtract, and rearrange equations to derive new knowledge. Additionally, we are able to identify what similar or different physical phenomena are in order to reason about relationships between physical quantities. Lastly, when observing physics, we intuitively understand that solutions must evolve forward with time and be temporally coherent. In the context of machine learning, these behaviors would have to be manifested in latent representations of physical solutions, and to this end, we demonstrate that masked autoencoders can learn expressive, general representations of PDEs.

To do this, we propose a set of experiments to compare the MAE model against other pretraining paradigms such as contrastive or transfer learning. Firstly, we pretrain and MAE or contrastive encoder (35) on the 1D KdV-Burgers pretraining set. Additionally, we pretrain a FNO and Unet model to predict future timesteps on this dataset in order to model a transfer learning scenario. Given these pretrained models, we seek to understand their latent representations. To do this, we embed PDE samples from three different scenarios.

Firstly, we embed equation samples from the Heat and Burgers equations and add these two embeddings in latent space; the average pairwise distance is calculated between this summed embedding and a corresponding embedding from the viscous Burgers equation, which is an experiment in latent arithmetic (Arithmetic). Next, the average pairwise distance between embeddings from the Heat equation is calculated; embedded samples have varying initial conditions but evolve with the same coefficients, which tests if models can embed similar dynamics to similar representations (Similarity). Lastly, he average pairwise distance between embeddings of subsequent timesteps of the Heat equation is calculated, measuring temporal coherence of latent embedding (Temporal). To ensure a fair comparison, embeddings are projected to a common dimension $d = 16$ with PCA and normalized ($v = v/max(||v||_2)$). Masked pretraining performs well across these experiments and learns general representations due to its minimal inductive bias, compared to contrastive learning or neural solvers which have specific objectives to either maximize similarity or predict next the timestep.

**Table 18:** Models are pretrained on the same KdV-B dataset and used to produce latent embeddings across different scenarios. The average pairwise L2 distance is reported in each case.

| Model | Arithmetic ↓ | Similarity ↓ | Temporal ↓ | Average ↓ |
|---|---|---|---|---|
| MAE | **0.5072** | 1.3674 | **0.5277** | **0.8007** |
| Contrastive | 1.4839 | **1.3131** | 0.8626 | 1.2199 |
| FNO | 0.9442 | 1.3562 | 0.7565 | 1.0190 |
| Unet | 0.9447 | 1.3562 | 0.7397 | 1.0135 |

