# OpenReview forum: "Masked Autoencoders are PDE Learners"
_TMLR — Accepted by TMLR_

### Review · Reviewer_9TVg · 2024-10-12

**Summary Of Contributions:**

The submission describes the results of adapting Masked Autoencoding Pretraining to the PDE setting. They test the approach across a variety of basic 1D and 2D PDEs and find somewhat mixed performance. The pretrained encoder effectively distinguishes between PDEs, and as consequence also effectively predicts their properties. It also does well on in-distribution downstream tasks but does poorly OOD. The submission also includes useful ablations on choices made for the pretraining model.

**Audience:**

Yes

**Broader Impact Concerns:**

No broader impact concerns.

**Claims And Evidence:**

No

**Requested Changes:**

Overall the breadth of the experiments is impressive and suggests there is content here meriting eventual acceptance, but the submission is not currently ready for publication. The main issues are the heavily truncated conference-style text and some factual errors around the differences between the PDEs.

__Critical__

1. Rewrite this into a journal paper - it's clear that the current submission is written to be a space-limited conference paper. Given the less restrictive page limits in TMLR, the authors should de-compress the submission significant. Architectural and patching decisions, what the setup is for super resolution or various tasks, and others should not require digging into the appendix. The final paper should be able to be read straight through with the user understanding all vital details. I'd really recommend using the space that's available. Define losses. Describe architectural choices. Include figures if it seems difficult to get points across in text.
2. It is fine to include the current tests as "in distribution" but transfer from KdV-Burgers to subsets of the same equation is not generalization testing. Either update the text and explore around parameter ranges or focus on true OOD tasks like KS and NS.
3. Label axes on figures. Especially given the limited diversity in the dynamics and initial conditions, it is honestly very hard to tell which direction is "forward" in time.
4. Tie more results to downstream tasks - This connects to the motivation issues. For the paper to be interesting, the results need to be tied to tasks that researchers are actually trying to solve. The current task selection is diverse, but many of these are toy tasks that previous works have not tried to address and there's nothing explaining why we should care about these tasks. For the results to merit inclusion in the main text, they need to be connected to a concrete goal or tell us something interesting about how the model works.

__Stronger paper__

1. The intro could use a pretty serious overhaul. Right now the motivation for this work is reads as "other people are doing things in this area". Strong motivation can make the paper significantly more interesting to readers who aren't finding this simply to look at MAE results for PDE data.
2. Find ways to include ablation-like content in the main text (along with 4 in critical requests). These are some of the more interesting results to readers who DO work in this space. This is a largely empirical paper - it is valuable for people working in similar spaces to see how your choices impacted downstream tasks.
3. Compare your results to other pretraining methods. While it's a new space, at this point there are 10+ "foundation models" for PDEs many of which have open source implementations and weights. For an empirical paper, it would be extremely helpful to readers to better understand how your approach compares to those on the tasks for which they're designed. It is not important for acceptance that the proposed method outperform them.
4. Improve the discussion of the results - a lot of the discussion of results right now focuses on what is in and out of distribution, but based on the notes on KdV-B, it does not seem to be accurate. The paper currently does a good job of documenting experiments, but it doesn't provide insight into what works or why.
5. The impact of training in "latent space evaluation" is currently unclear. There are no baselines. Does the raw "pixel-space" data also cluster in this way? Without such a baseline, we cannot tell if this clustering of the autoencoder result is useful. For instance, if we found that pixel-space results could separate Dirichlet and Neuman BCs then this would indicate that the model is discarding important information.
6. Focus more on 2D and extend to 3D. 1D results lack impact. Numerical solvers for 1D PDEs are extremely fast to the point where solver speed for 1D PDEs is only a bottleneck for special cases like stiff systems. On top of this, very few real-world problems can be approximated accurately in 1D. There are even many documented cases where the same equations behave differently in different dimensions (turbulence in Navier-Stokes being the classic example here). 1D are often used for testing very specific phenomena - wave-front tracking for instance, but general prediction accuracy in 1D isn't that interesting unless you are specifically highlighting areas numerical methods do poorly.

**Strengths And Weaknesses:**

__Strengths__

1. The effort to adapt pretraining methods to PDEs is timely and seems like a reasonable idea.
2. The breadth of experiments are impressive and creative.
3. Claims are fairly measured and do not overstate the results outside of the OOD claims discussed below. Areas where the method performed poorly are shown clearly.

__Weaknesses__
1. This is currently not written for a journal format. Understanding the specifics of the paper requires significant jumps between the main body and the appendix for details. A journal submission should contain all required information to understand precisely what was done and why. Particular technical details or extra experiments that are not core to the paper's goals can be left to the appendix, but simple information like patching approaches or PDE parameter ranges are core to the understanding the paper content.
2. The paper currently lacks motivation. There is never any overarching goal mentioned or real world problem looking to be addressed besides that other ML researchers are working in the space yet comparisons to those other works are highly limited. This is sufficient for other members of that community who are already familiar with what their goals are, but a reader currently does not come away from this understanding why individuals are using ML to solve PDEs. Some specific examples:
      1. _"any important advances have been made to address these challenges, with SOTA models outperforming numerical solvers on well-studied PDEs under certain setups, proposing error bounds, and being extended to solve real-world problems"_- This is very vague. How are they outperforming numerical methods? Are they faster? Are they more accurate? Have they been able to be used on new problems that numerical methods didn't apply to? Note that a number of the citations here have been referenced as unfair numerical comparisons in [recent work](https://www.arxiv.org/pdf/2407.07218) discussing baselines used in this space. The stated goals of the papers can certainly be used to motivate more work in the space, but we should be cautious when stating that one method has outperformed others.
      2. Many of the experiments come across as arbitrary. They do not seem tied to a real problem someone who might use your method might have. For instance, ablation results are typically listed in terms of reconstruction error - this is a synthetic task used to avoid requiring labels. The result that would be interesting to readers is downstream performance.
      3. However, downstream tasks also suffer from this issue. The addition of two latent states, for example, even with a background in this area, I cannot really see a case where this would be practically useful even if the the target task was not a subset of the training data (addressed below). This also applies to the resolution prediction tasks - users know the resolution of their data. It actually seems somewhat troubling that the model is able to recover this data and something that should discussed.
3. The submission seems to use in and out of distribution labels inaccurately. For example, if we look at 1D data, the training set is KdV-B. This is a superset of the KdV, Burgers, and heat equation. Burgers is a non-linear extension of 1D advection. The gap between advection and the training data seems to be the rate of advective transport in the coefficient sampling rather than the equation itself.
4. The resulting 1D visualizations all seem quite similar. Because the axes aren't labeled, it's difficult to determine the exact pattern, but there seems to be very little transport across the domain in any of the examples besides 1D advection and KS.
5. Figures and tables are at times underlabeled. In figure 2, one can't see which direction is forward in time. The colors make two losses in one table readable, but splitting them out might make it clearer. However, table 3 just uses different losses in different rows.
6. Terminology is sometimes inaccurate. For instance "Heat-Burgers" is just viscous Burgers - a term already used elsewhere. Dirichlet and Neumann BCs are entire families of BCs rather than specific BCs. The specific families are included in the appendix, but this should be made clear in the main text.

---

> ### Author Response · Authors · 2024-10-18
> **Response to Reviewer 9TVg**
>
> Thank you for the thorough review! From your comments, we've changed quite a lot about the writing and format of the paper, which we've highlighted in blue in the updated manuscript. To summarize our changes according to the feedback:
>
> ### 1. Rewriting into a journal paper (Critical Change 1, Weakness 1)
> Totally valid, we agree with this sentiment and we have decompressed much of the content and included information from the Appendix into the main body. In particular, many details about the methods, data augmentations, PDE datasets, and training tasks were added to the main body. Furthermore, we have gone through and revised many portions of the paper to improve clarity.
>
> ### 2. Imprecise wording around generalization testing (Critical Change 2, Weakness 3/6)
> This was an oversight on our part. We have updated the text to reflect that transferring between KdV-Burgers and the Heat/Burgers equations is not generalization testing. We have also tried to be more precise with the language surrounding in and out of distribution samples. Furthermore, we've updated the terminology about the viscous Burgers equation and described the specific Dirichlet and Neumann conditions in the main body.
>
> ### 3. Labeling figures (Critical Change 3, Weakness 4/5)
> Thank you for pointing this out! We have updated those figures with labeled axes to denote space and time within the figure.
>
> ### 4. Additional motivation surrounding results (Critical Change 4, Weakness 2.2/2.3)
> We have added an additional few paragraphs describing the motivation for running these experiments in Section 5, as well as reorganized our results section to make this more clear.
>
> ### 5. Additional motivation for the study (Stronger Change 1, Weakness 2.1)
> Within the introduction, we have added an additional paragraph describing why the study uses a masked modeling approach and why this could be an effective method. While, empirically through experiments there are some clear limitations, the original intent of the study had motivation stemming from the broader ML community.
>
> ### 6. Including ablation-like content in the main body (Stronger Change 2)
> This is a great suggestion. We have included some of this content, such as the Lie augmentation and multi-resolution ablations, into the main body which has improved the readability of our paper.
>
> ### 7. Situating results within other pretraining/foundation model works (Stronger Change 3)
> This is a tough one, since it would require running an entirely new set of experiments. We are willing to admit that the proposed approach will likely not outperform these foundation models in timestepping tasks, however, the idea behind our paper is somewhat more nuanced. Encoder-style approaches, such as [1] and [2], tend to not benchmark against foundation models, since PDE encoders tend to be more flexible (i.e., applied to any downstream architecture, applicable to different tasks). However, the true utility of this is unclear and seems to be largely under-explored in this field. Perhaps future work will find an interesting set of uses for PDE encoders. In any case, we have tried to be transparent and added a discussion about this in Section 6.
> - 1. Grégoire Mialon, Quentin Garrido, Hannah Lawrence, Danyal Rehman, Yann LeCun, Bobak T. Kiani. Self-Supervised Learning with Lie Symmetries for Partial Differential Equations. 2023.
> - 2. Rui Zhang, Qi Meng, Zhi-Ming Ma. Deciphering and integrating invariants for neural operator learning with various physical mechanisms. 2024.
>
> ### 8. Additional discussion about the proposed method (Stronger Change 4)
> Thank you for the suggestion! We have created an additional discussion section (Section 6) and added some details there.
>
> ### 9. Comparing latent space evaluation to baselines (Stronger Change 5)
> This is a completely valid point and we have updated the t-SNE plots to show a baseline of clustering raw data where we found no noticeable trends. We believe that the raw data may be too high dimensional for dimensionality reduction techniques alone to find trends, the use of a pretrained encoder can accentuate differences in samples that are physically meaningful.
>
> ### 10. Focusing on 2D and 3D systems (Stronger Change 6)
> We agree with the comment and this would definitely more impactful, however, for the present study it seems to be out of scope given the short revision timeline. We have noted this as a direction for future study.

---

> > ### Comment · Action_Editor_5Ldh · 2024-11-04
> >
> > Dear reviewer 9TVg,
> >
> > The authors have provided a response to your review.
> >
> > When is convenient to you, please read this response and the revised manuscript. If you have more questions, please reach out to the authors. Otherwise, please submit your recommendation on the status of this paper.
> >
> > Best,
> >
> > AE

---

### Review · Reviewer_ejJu · 2024-10-12

**Summary Of Contributions:**

This paper proposes an innovative approach using Masked Autoencoders (MAEs) for solving Partial Differential Equations (PDEs). The authors adapt self-supervised masked pretraining, a method typically used in natural language processing and computer vision, to physics problems. The primary contributions include demonstrating that MAEs can effectively generalize to unseen PDEs, improve performance in downstream tasks like PDE coefficient regression, and enhance the super-resolution of neural solvers. The authors present a framework for latent PDE arithmetic, showing that learned representations from MAEs are meaningful and can operate in a latent space.

**Audience:**

Yes

**Claims And Evidence:**

Yes

**Requested Changes:**

Major concerns:

1. Provide More movitation and Additional Quantitative Results for Latent Arithmetic:

What is the rationale of doing latent arithmetic in the context of learning PDE solutions? Although viscous Burgers Eqn can be treated as the linear combination of invicid Burgers and Heat Eqn, it does not necessarily mean that the same holds in the latent space, mainly because of the nonlinearity of MAE and the PDE. The author should provide some justification/motivation of the latent arithmetics.
Page 7 bottom: the author claim “MAE encoer/decoder has never seen the heat/burgers eqn” is this correct? Shouldn’t the MAE be pre-trained on kdv-burgers? In Appendix B.1, alpha, beta, and gamma are allowed to be zero so KDV-Burger actually contains heat and burgers equation.

Including a quantitative comparison between reconstructed solutions from latent arithmetic and actual PDE solutions could strengthen the evidence for this method's effectiveness. For example, could you include viscous Burgers in Figure 4C and see if the cluster lie in the space spanned by the clusters of invicid Burgers and Heat Eqn?

2. Justification for classification of BC:

My main concern is that classifying the Boundary condition from PDE solution data is not a well-posed problem. For a fixed PDE solution, one can calculate its Dirichlet boundary data and Neumann boundary data and in general Robin boundary data. What is point to infer boundary condition type for the PDE data? As the author pointed out in Figure 4 E, Solutions generated from Dirichlet and Neumann BC are mixed. This is quite natural because a Dirichlet BC f and a different Neumann BC g may generate the same PDE solution u.

3. Discuss Computational Requirements:

A section on the computational costs associated with training MAEs on multi-resolution and diverse PDE data would be helpful for readers considering implementation.

4. Introduce Time-stepping and Super-Resolution:

A brief introduction on the problem setup and main goals of time-stepping and super-resolution should be provided.


Minor issues:

No year in reference [54]

The term Heat-Burgers PDE is very weird. It should be called viscous Burgers equation.

The last sentece in the paragraph above section 4.4 is not clear.

Section 4.5, reference [64] was not shown correctly.

**Strengths And Weaknesses:**

Strengths:

Novelty: The adaptation of masked pretraining to the domain of PDEs is innovative, expanding the potential applications of this self-supervised learning method.

Comprehensive Evaluation: The authors evaluate the approach on a range of PDEs, including previously unseen equations, and cover diverse tasks such as time-stepping and super-resolution.

Practical Implications: Demonstrating improvements in both time-stepping accuracy and super-resolution highlights the potential utility of MAEs in real-world physics applications.


Weakness:

Limited Extrapolation: Although the MAE model shows generalization to some unseen PDEs, performance declines significantly when applied to more complex equations like the Navier-Stokes equations.

Ambiguity in Latent Arithmetic Results: While latent PDE arithmetic is an exciting idea, the paper does not clearly quantify how closely the latent solutions match true solutions.

---

> ### Author Response · Authors · 2024-10-18
> **Response to Reviewer ejJu**
>
> Thank you for spending the time to read and review our manuscript. We've made some changes to address these comments, which we've highlighted in blue in the updated manuscript. To summarize our changes:
>
> ### 1. Latent PDE arithmetic results (Major Concern 1)
> Thank you for raising this concern. We had some discussion about this and decided that we would reduce our paper’s claims, move the latent arithmetic results to the appendix and mention it as a direction for future work. It seems like an interesting research topic, but the current work is already quite large in scope and latent arithmetic may be best left for another study.
>
> ### 2. Classification of boundary conditions (Major Concern 2)
> Thank you for raising this point, indeed you are correct that there is not much practical value in inferring BCs since it is already known. Indeed, much of the coefficient regression and PDE feature prediction was done mainly to test the MAE’s learned representations, much like visualizing its latent space. Regressing simple values that are known a priori is much like using a linear probe in CV contrastive self-supervised learning; one example would be regressing the rotation of an image or if it is grayscale or not, which are both known from the data augmentation process. We have added a discussion to clarify this in our new Results section (Section 5), highlighted in blue.
>
> ### 3. Computational requirements (Major Concern 3)
> We have added these details in Appendix A, highlighted in blue. Thanks for the suggestion!
>
> ### 4. Additional details on time-stepping and super-resolution (Major Concern 4)
> Appreciate the suggestion, we have added details about this in Section 5.4, highlighted in blue.
>
> ### 5. Minor concerns
> Minor concerns 1, 2, and 4 have been fixed, the discussion about 3 has been removed to improve clarity. Thanks!
>
> Accidentally posted one of the other responses to this thread, so I deleted it. Apologies for that!

---

> ### Comment · Action_Editor_5Ldh · 2024-11-04
>
> Dear reviewer ejJu,
>
> The authors have provided a response to your review.
>
> When is convenient to you, please read this response and the revised manuscript. If you have more questions, please reach out to the authors. Otherwise, please submit your recommendation on the status of this paper.
>
> Best,
>
> AE

---

### Review · Reviewer_rzub · 2024-10-13

**Summary Of Contributions:**

This work explores the use of masked autoencoders (MAEs) for learning representations of partial differential equations (PDEs). The key idea is to adapt the successful MAE approach from computer vision to the domain of PDEs, with the goal of learning generalizable representations that can be useful for various PDE-related downstream tasks. In particular, the authors pretrain MAE models on datasets of solutions to various PDEs (KdV-Burgers in 1d and Heat, Advection, and Burgers in 2d), and then evaluate the learned representations on tasks like PDE feature prediction, time-stepping, and super-resolution across different equations. In general, the experiments show a positive impact of pretraining, in particular in combination with additional fine-tuning, and a benefit over contrastive approaches in out-of-distribution settings.

**Audience:**

Yes

**Claims And Evidence:**

No

**Requested Changes:**

1. *(critical)* How sensitive is the method to hyperparameter choices (see "Performance" above)?

2. *(critical)* How does the performance compare to various other pretraining approaches beyond the contrastive learning baseline (see "Baselines" above)?

3. How well does this approach scale to much larger and more complex PDE datasets (see "Datasets" above)?

4. *(critical)* Can the latent PDE arithmetic be quantitatively evaluated (see "Tasks" above)?

5. Can one investigate the theoretical properties of the proposed method (see "Adaptation to PDEs")?

**Strengths And Weaknesses:**

### Strengths

- Novel application of a successful self-supervised learning technique (ViT-based MAEs) to PDEs.

- Positive empirical results showing benefits of MAE pretraining and finetuning for various PDE types and downstream tasks.

### Weaknesses

1. **Performance**: Modest performance gains compared to baselines.

    - Does not consistently outperform simpler approaches like conditioning on ground truth coefficients (at least in 1d).

    - Mostly requires retraining the full encoder for best performance on downstream tasks, incurring significant additional costs.

    - Limited ability to extrapolate to PDEs very different from those seen during pretraining, not sufficiently supporting statements such as "Through self-supervised learning, MAE models can learn trends in PDEs without labeled data and in equations outside the training set.", "We show that learned representations can generalize to unseen equations or parameters".

    - Not clear how sensitive the results are w.r.t. hyperparameter choices like architectures, training methods, masking ratio, etc. For instance, using a randomly initialized and fine-tuned ViT encoder almost performs as well as the finetuned MAE (in particular in 1d), indicating that the benefits might be due to a higher parameter count or architectural prior instead of the pretraining strategy.

2. **Baselines**: Insufficient comparisons against other baselines.

    - Another baseline should be to train different models (such as UNet, FNO, and Transformer NOs) directly for time-stepping on the same pretraining data used by the MAE. The trained models can then be used for the same downstream tasks, potentially after finetuning.

    - Comparisons against other approaches of pretraining foundation models are missing, e.g., MPP (https://arxiv.org/abs/2310.02994), DPOT (https://arxiv.org/pdf/2403.03542), CoDA-NO (https://arxiv.org/abs/2403.12553), Poseidon (https://arxiv.org/pdf/2405.19101), Strategies for Pretraining Neural Operators (https://arxiv.org/abs/2406.08473), which all appeared at least 3 months before the submission. In particular, CoDA-NO explores masking strategies similar to MAEs. Also, only a small comparison against the contrastive SSL approach by Mialon et al. (2023) is presented.

    - Why is only a ViT baseline considered for feature prediction tasks and there are no comparisons against other (e.g., simpler) models? Details on training the baseline and finetuning the MAE seem to be missing, however, are crucial for a fair comparison.

    - Training and inference costs are not reported and compared. In particular, the higher parameter count and need for additional finetuning might make the method less competitive against baselines and numerical solvers.

    - One should compare directly to FNOs native superresolutions capabilities. In particular, it is not clear why the ResNet and the interpolation are necessary. For reference, one should also show the performance of only using interpolation.


3. **Datasets**: Limited evidence for scalability to larger and more complex PDE datasets.

    - The paper focuses mainly on relatively simple 1D and 2D PDEs and the findings may not generalize to more complex systems. In particular, on more complex equations (such as NS) or out-of-distribution datasets performance seems to be limited. For instance, the t-SNE plots seem less structured for the 2d equations and it is also noted that "trends in 2D PDE data are more difficult to learn without labels".

    - Previous models have considered pretraining on a much larger scale and with PDEs that are practically more relevant (see the datasets used in the pretraining foundation models mentioned above). Such scaling also seems necessary to verify the claim that "With enough PDE data during pretraining, freezing MAE encoders can be more practical since the pretraining distribution would cover most downstream cases".


3. **Tasks:** Artificial downstream tasks and preliminary latent PDE arithmetic.

    - Some downstream tasks seem quite artificial and not of practical interest, such as classifying boundary conditions or predicting resolutions (which are also both easily done from the input).

    - Results on latent PDE arithmetic are very preliminary and it is unclear if it can be used for meaningful applications. There seems to be only one qualitative Figure with a single example to validate the latent PDE arithmetic. This is not sufficient to support statements like "learned representations [...] are semantically meaningful by performing latent PDE arithmetic." or "Additionally, we propose operating in this latent space to interpolate between encoded PDE vectors".


4. **Adaptation to PDEs**: Limited adaptation and theoretical analysis of MAEs in the context of PDEs.

    - It seems like the paper is coming from a computer vision perspective without many PDE-specific adaptations. While this is generally not a shortcoming, it is unclear what is meant by the "several PDE-specific modifications" used for pretraining. Discretization-invariance is claimed as an important feature but ViTs are used as the backbone which despite the tweaks mentioned in "Multi-Resolution pretraining" cannot be consistently applied to arbitrary resolutions. In particular, there already exist more principled approaches to designing discretization-invariant ViTs (https://arxiv.org/abs/2406.06486).

    - It is not investigated how physically valid masking is. For instance, is there even a unique solution for the considered PDEs or will the model regress the mean?

    - It should be explained what is meant by "labels" in the context of PDEs, e.g., in "unlabeled, and heterogeneous datasets to learn latent physics at scale" and "can learn highly structured latent spaces without the need for labels". For instance, for a time-stepping objective, the same data as used for MAE is sufficient and no labels seem to be needed (see also "Baselines" above). Also, it is not clear what effort is meant by "so no additional effort is needed to generate pretraining labels". Typical labels, such as the coefficients, boundary conditions, etc., are anyways given when obtaining the data from a numerical solver.

### Minor issues and questions

1. It's not clear why the 1d Heat and Burgers eqs. (with Dirichlet boundary condition) are not "explicitly seen" in the 1d KdV-Burgers eq.? They seem to be at least in the training distribution.

2. It should be explained what is meant by "PDE dynamics information"?

3. What is an "output functional space"?

4. "We hypothesize that the increased difficulty of the task increases the importance of MAE encoder guidance in time-stepping and super-resolution. This increased difficulty also manifests itself when extrapolating pretrained encoders to new PDEs, as the performance on the Navier-Stokes equations is limited." This sounds like a contradiction, since the MAE encoder guidance does *not* help for NS?

5. It is not clear what the purpose of the smoke buoyancy experiment is.

6. Why would we even want different resolutions to be mapped to different parts of the latent space as shown in Figure 4D?

7. Code is not yet available.

---

> ### Author Response · Authors · 2024-10-18
> **Response to Reviewer rzub**
>
> Thank you for the detailed and comprehensive review! Your extensive knowledge and suggestions have helped to improve our manuscript. We've made some changes to address your comments, which we've highlighted in blue in the updated manuscript. To summarize our changes:
>
> ### 1. Hyperparameter sensitivity (Requested Change 1, Weakness 1)
> Thank you for raising this concern. Based on our internal experiments, we have identified that a ViT is a preferred architecture as an autoencoder due to its tokenization. FNO autoencoders tend to have poor reconstructions due to introducing spurious high frequency modes when masking spatially. Unet approaches fare somewhat better but still suffer sharp boundaries across masked and unmasked regions. Additionally, we evaluated different ViT variants, such as ViViT (axial attention) and Swin Transformer (window attention). We found that they performed worse due to restricting the attention mechanism, and the additional speedups were not significant due to reducing compute when masking out large portions of the input.
>
> Additionally, we did evaluate different masking ratios based on coefficient regression accuracy, and empirically chose 75% and 90% for 1D and 2D PDEs, however this was not extremely sensitive and ratios +-10% would be reasonable. Most other hyperparameters vary smoothly, for example reducing patch size slightly increases performance across downstream tasks. In general, the model did not seem to be overly sensitive to hyperparameters, and we have added some discussion about these points in Appendix A.
>
> ### 2. Performance to other baselines (Requested Change 2, Weakness 2)
> This is a completely valid criticism, and we are willing to admit that the proposed approach will likely not outperform these foundation models in timestepping tasks. Given the short revision timeline, it would be challenging to properly verify this claim, but there is some precedent from previous work that supports this [1]. However, the idea behind our paper is somewhat more nuanced. Encoder-style approaches, such as [2] and [3], tend to not benchmark against foundation models, since PDE encoders tend to be more flexible (i.e., applied to any downstream architecture, applicable to different tasks). However, the true utility of this is unclear and seems to be largely under-explored in this field. Perhaps future work will find an interesting set of uses for PDE encoders. In any case, we have tried to be transparent and added a discussion about this in Section 6.
> - 1. Anthony Zhou, Cooper Lorsung, AmirPouya Hemmasian, Amir Barati Farimani. Strategies for Pretraining Neural Operators. 2024.
> - 2. Grégoire Mialon, Quentin Garrido, Hannah Lawrence, Danyal Rehman, Yann LeCun, Bobak T. Kiani. Self-Supervised Learning with Lie Symmetries for Partial Differential Equations. 2023.
> - 3. Rui Zhang, Qi Meng, Zhi-Ming Ma. Deciphering and integrating invariants for neural operator learning with various physical mechanisms. 2024.
>
> ### 3. Scaling to larger datasets (Requested Change 3)
> This is a great suggestion, and we would also be curious to know what the scaling behavior would be like. However, given the short revision timeline, it seems to be out of scope for now and we have added this as a direction for future work.
>
> ### 4. Latent arithmetic evaluation (Requested Change 4)
> Thank you for raising this concern. We had some discussion about this and decided that we would reduce our paper’s claims, move the latent arithmetic results to the appendix and mention it as a direction for future work. It seems like an interesting research topic, but the current work is already quite large in scope and latent arithmetic may be best left for another study.
>
> ### 5. Theoretical properties (Requested Change 5)
> Thank you for the suggestion, we have added a new discussion section (Section 6) and provided some insight about the results and method.

---

> > ### Author Response · Authors · 2024-10-18
> > **Response to Reviewer rzub**
> >
> > ### 6. Minor issues and questions
> > > It's not clear why the 1d Heat and Burgers eqs. are not "explicitly seen" in the 1d KdV-Burgers eq.?
> >
> > Thanks for pointing this out, this is an oversight on our part. We have updated the language to reflect the fact that the Heat and Burgers equations are within the training distribution.
> >
> > > It should be explained what is meant by "PDE dynamics information"?
> >
> > Apologies for the unclear language, the meaning we were going for is “time-dependent trends”. For example, the advection equation has different dynamics than the heat equation, since it tends to move waves in time vs. diffusing them. We’ve added a note about this, thanks!
> >
> > > What is an "output functional space"?
> >
> > Thanks for pointing this out, a bit of sloppy language on our part. We have corrected it to be a function space, which is a topological space whose points are functions. One example is the set of all functions that map real numbers to real numbers.
> >
> > > This sounds like a contradiction, since the MAE encoder guidance does not help for NS?
> >
> > This is a great question. Not sure what we were going for there, but we’ve changed the wording to: "We hypothesize that the increased difficulty of the task increases the importance of MAE encoder guidance in time-stepping and super-resolution. However, out-of-distribution datasets are still challenging: when extrapolating pretrained encoders to new PDEs, such as the Navier-Stokes equations, the performance is limited."
> >
> > > It is not clear what the purpose of the smoke buoyancy experiment is.
> >
> > This is a valid point; we were mostly curious if the MAE strategy would work on more complex 2D phenomena. There isn’t a clear takeaway, but if people are curious to apply the model to more complex phenomena, it could be a starting point.
> >
> > > Why would we even want different resolutions to be mapped to different parts of the latent space as shown in Figure 4D?
> >
> > This is an interesting question. There could be arguments either way, but in general we believe that the model should be able to discern different resolutions. One could say that different resolutions still model the same phenomena, therefore the latent representation should be the same. However, different resolutions would need to be treated differently internally by a transformer, since they have different numbers of tokens, therefore the hidden representation should benefit by being distinct.
> >
> > > Code is not yet available.
> >
> > Code will be released with the camera ready version, due to anonymity reasons it has been omitted.

---

> ### Author Response · Authors · 2024-10-18
> **Response to Reviewer rzub**
>
> ### 7. Clarifications to some weaknesses
> > Limited ability to extrapolate to PDEs very different from those seen during pretraining
>
> This is a fair criticism and we will reduce our claims. We have tried to be transparent by including all results and will add some additional discussion about the negative results in limitations.
>
> > Using a randomly initialized and fine-tuned ViT encoder almost performs as well as the finetuned MAE (in particular in 1d), indicating that the benefits might be due to a higher parameter count or architectural prior instead of the pretraining strategy.
>
> This is true. However, we note that trends in 1D tend to be much easier to learn, which might explain why random initialization is able to perform well. In 2D, the pretrained encoder exhibits more benefits, which seems to support the utility and purpose of using the proposed method.
>
> > Why is only a ViT baseline considered for feature prediction tasks and there are no comparisons against other (e.g., simpler) models?
>
> This is a fair criticism, but the feature prediction task is generally meant to elucidate what the ViT model has learned during pretraining, not to actually regress coefficients. Comparing against a supervised baseline of the same architecture seems to be a common approach, such as in Table 1 in [1] or Table 7 in [2]
> - 1. Grégoire Mialon, Quentin Garrido, Hannah Lawrence, Danyal Rehman, Yann LeCun, Bobak T. Kiani. Self-Supervised Learning with Lie Symmetries for Partial Differential Equations. 2023.
> - 2. Zhenda Xie, Zheng Zhang, Yue Cao, Yutong Lin, Jianmin Bao, Zhuliang Yao, Qi Dai, Han Hu. SimMIM: A Simple Framework for Masked Image Modeling. 2021.
>
> > Higher parameter count and need for additional finetuning might make the method less competitive against baselines and numerical solvers.
>
> This is true, and we have added a comment pointing out that full fine-tuning is likely not a feasible approach in the limitations.
>
> > One should compare directly to FNOs native superresolutions capabilities. In particular, it is not clear why the ResNet and the interpolation are necessary.
>
> Thank you for raising this concern. We use this specific SR implementation due to a previous study demonstrating its benefits [1]. This study also compares to FNO as well as bicubic interpolation, and finds that their method is better (Appendix, Table 2 of their paper). We hope that this can give some additional insight.
> - 1. Qidong Yang, Alex Hernandez-Garcia, Paula Harder, Venkatesh Ramesh, Prasanna Sattegeri, Daniela Szwarcman, Campbell D. Watson, David Rolnick. Fourier Neural Operators for Arbitrary Resolution Climate Data Downscaling. 2023.
>
> > Some downstream tasks seem quite artificial and not of practical interest, such as classifying boundary conditions or predicting resolutions (which are also both easily done from the input).
>
> This is completely correct. We recognize that coefficient regression is an artificial task and doesn’t have much practical interest, but we introduced PDE feature prediction mainly to test the MAE’s learned representations. Regressing simple values that are known a priori is much like using a linear probe in CV contrastive self-supervised learning; one example would be regressing the rotation of an image or if it is grayscale or not, which are both known from the data augmentation process. We have added a discussion to clarify this in our new Results section (Section 5), highlighted in blue.
>
> > It is not investigated how physically valid masking is.
>
> Thank you for this suggestion! We have added a discussion about masking in Section 6.
>
> > It should be explained what is meant by "labels" in the context of PDEs.
>
> This is a great suggestion, we have modified the manuscript to more faithfully reflect the fact that labels are already given alongside the PDE data in most PDE tasks.

---

> > ### Comment · Reviewer_rzub · 2024-10-28
> >
> > I thank the authors for their revision and response, clarifying parts of my questions and fixing most minor issues. Also, I appreciate the transparency in terms of limitations and agree that some of the proposed changes might require more time. However, the paper's contribution seems rather incremental without these changes, given the current response. For instance, after the response, it seems clear that the paper does not provide *significant* contributions in terms of the following points:
> > - Comparisons to baselines and practically of the proposed approach (e.g., given the requirement for additional fine-tuning and the availability of labels for most PDE tasks),
> > - Latent arithmetic,
> > - Smoke buoyancy experiments or more complex 2D/3D problems,
> > - Clear benefits over simpler baselines and for extrapolating to unseen equations or parameters.
> >
> > ---
> >
> > Let me comment on some of the items:
> >
> > 1. **Hyperparameter sensitivity**
> >
> >     Thank you for adding additional discussion. It makes sense that FNOs are unsuitable for spatial patching due to discontinuities leading to Gibbs phenomena. However, it seems that most of the new claims (in Appendix A and the response) are made without any empirical evidence.
> >
> > 2. **Performance to other baselines**
> >
> >     I appreciate the transparent and honest discussion, e.g., "While this versatility is interesting, the practicality of this is unclear".
> >
> >     However, this raises an important question: Are these approaches really more "versatile" since one could also use hidden representations of foundation models for these tasks? Without numerical evidence for this claim and given the unclear "practicality" of the proposed approach, it seems that the contribution of the present paper is rather incremental.
> >
> > 6. **Minor issues and questions**
> >
> >     > Code will be released with the camera ready version, due to anonymity reasons it has been omitted.
> >
> >     Since one could anonymize the code, this seems not to be a valid argument. However, I appreciate that the code would be released with the camera ready version.
> >
> >     > However, different resolutions would need to be treated differently internally by a transformer, since they have different numbers of tokens, therefore the hidden representation should benefit by being distinct.
> >
> >     To me, this seems more like an artifact of using transformers instead of neural operators, and I wonder what the practical use case for this task would be (which is also connected to the general "practicality" as discussed above).
> >
> > 7. **Clarifications to some weaknesses**
> >
> >     > This is a fair criticism, but the feature prediction task is generally meant to elucidate what the ViT model has learned during pretraining, not to actually regress coefficients. Comparing against a supervised baseline of the same architecture seems to be a common approach, such as in Table 1 in [1] or Table 7 in [2].
> >
> >     I know that this is a common approach, and I also think that it is good to include these results. However, it would be interesting to evaluate a baseline for directly regressing the coefficients to assess the task's difficulty.
> >
> >
> >     > Thank you for raising this concern. We use this specific SR implementation due to a previous study demonstrating its benefits [1]. This study also compares to FNO as well as bicubic interpolation, and finds that their method is better (Appendix, Table 2 of their paper). We hope that this can give some additional insight.
> >
> >     Similar to the item above, I appreciate the baseline, but I think another, arguably simpler, baseline would be interesting.
> >
> >     > This is completely correct. We recognize that coefficient regression is an artificial task and doesn’t have much practical interest, but we introduced PDE feature prediction mainly to test the MAE’s learned representations. Regressing simple values that are known a priori is much like using a linear probe in CV contrastive self-supervised learning; one example would be regressing the rotation of an image or if it is grayscale or not, which are both known from the data augmentation process. We have added a discussion to clarify this in our new Results section (Section 5), highlighted in blue.
> >
> >     This brings us back to the "practicality" of the proposed approach, a question that definitely has a positive answer in CV.

---

> > > ### Author Response · Authors · 2024-11-04
> > > **Response to Reviewer rzub**
> > >
> > > Thank you for the reply and for taking the time to read through and give feedback on the revisions. We have taken your advice and run additional baselines and experiments, described below:
> > > ### 1. Simple baselines for Feature Prediction
> > >
> > > We have implemented some simple CNN and MLP baselines for regressing/classifying PDE features to judge the difficulty of these tasks, in Table 3 and 4 in the updated manuscript. Thank you for the suggestion!
> > >
> > > ### 2. Simple baselines for Super-Resolution
> > > We have also implemented some simple interpolation baselines for the super-resolution experiments to give some additional insights to the model's performance, in Table 5 and 6 of the updated manuscript.
> > >
> > > ### 3. Discussion on Versatility
> > > An interesting and worthwhile point on the model's contribution was raised:
> > > >Are these approaches really more "versatile" since one could also use hidden representations of foundation models for these tasks?
> > >
> > > We gave this some thought and wanted to conceptualize an experiment could tell us about the hidden representations of our model compared to other approaches. Testing "versatility" is not well-defined, but in the end we came up with a set of preliminary experiments to evaluate this and present the results in Appendix F in the updated manuscript. We hope that this can give some more insight into the current work.
> > >
> > > As a broader remark, we would agree that while the work is incremental, we hope that the extensive experiments and transparency can still be of benefit to some practitioners in the field. In any case, your responses have helped us to improve the work as well as contribute more insight to interesting problems, which we greatly appreciate.

---

### Decision · Action_Editor_5Ldh · 2024-11-18

**Recommendation:** Accept with minor revision

**Comment:**

While there were 2 leaning reject recommendations, these appeared to be grounded in a feeling that the amount of novelty was lacking and that there was not enough extensive experimentation. I agree these are limitations, but they do not meet the criteria for rejection from TMLR. The third reviewer, who recommended leaning accept, noted that the paper satisfied both criteria of acceptance by TMLR. In going over the reviewers and authors discussion, I find myself agreeing that both evidence and audience requirements are satisfied. Therefore, I recommend accept. Given that third reviewer noted several additional points (see below) that could be addressed by the authors to further strengthen their paper, I am recommending accept with minor revision, so that the authors can address these points.
-----------------------------------------
(Additional points from Reviewer 3):
1. Still no direct comparisons to other pretraining approaches/foundation models so it's unclear how MAE compares to these in any task.
2. "However, encoder-style approaches such as this work or contrastive PDE encoders (Mialon et al., 2023; Zhang et al., 2023) are more flexible than foundation models, capable of being applied to arbitrary downstream architectures and different fine-tuning tasks." does not really have any backing evidence since there are not comparisons.
3. In general, statements about how MAE is applicable to all data types are undercut when the paper uses augmentation strategies that require knowledge of the specific equations and which Lie symmetries apply. I don't believe MAE requires these so the statement is still valid, but the particular approach used in this paper may not generalize blindly.

The first two are the more notable ones, but while ultimately addressing them would make the paper far stronger and significantly more interesting, what is here does currently meet the two major criteria.

**Audience:**

All three reviewers agreed that the paper is of sufficient audience to the TMLR community. In addition, several reviewers noted that this is an area of growing research. For these reasons, I believe that at least some individual in TMLR's audience would be interested in knowing the authors' findings.

**Claims And Evidence:**

Two of the three reviewers - in their initial review or subsequent response - have said that the claims made were supported by evidence. The authors provided extensive responses to the third review who did not say that the claims were supported by evidence, including additional experiments after the reviewer submitted their review. Taking this all into account, I believe the authors did support their claims with sufficient evidence.

---

> ### Author Response · Authors · 2024-12-05
> **Thank you for the decision**
>
> Dear AC,
>
> Thank you for the recommendation and for the guidance on the additional points. We have uploaded our camera-ready version and discuss these additional points, however, feel free to reach out if there are any more concerns.
>
> 1. We have added a table in Appendix E.3 to compare the masked pretraining approach to a contrastive pretraining approach.
> 2. We have added a section in Appendix F to discuss how the latent embeddings generated by MAE models can be used in different contexts when compared to latent embeddings from contrastive or transfer learning baselines. We present some preliminary results suggesting that these embeddings encode more general/flexible information about the physical system, likely due to the masked learning objective having very little inductive bias, although we recognize that this is a challenging idea to rigorously test.
> 3. While Lie symmetries are not generally applicable, the data augmentation was mainly used to improve reconstruction accuracies slightly. For more general applications, these can be omitted; alternatively very simple augmentations, such as the ones tested the paper, can still be applicable, for instance shifting solutions in space (which most PDEs are invariant to).
>
> Thank you for your time and appreciate the effort from all the reviewers as well!